# Communication-Efficient Distributed Blockwise Momentum SGD with Error-Feedback

**Shuai Zheng**[*1,2]**, Ziyue Huang**[1]**, James T. Kwok**[1]

shzheng@amazon.com, {zhuangbq, jamesk}@cse.ust.hk

[1]Department of Computer Science and Engineering
Hong Kong University of Science and Technology
[2]Amazon Web Services

## Abstract

Communication overhead is a major bottleneck hampering the scalability of distributed machine learning systems. Recently, there has been a surge of interest in using gradient compression to improve the communication efficiency of distributed neural network training. Using 1-bit quantization, signSGD with majority vote achieves a 32x reduction on communication cost. However, its convergence is based on unrealistic assumptions and can diverge in practice. In this paper, we propose a general distributed compressed SGD with Nesterov's momentum. We consider two-way compression, which compresses the gradients both to and from workers. Convergence analysis on nonconvex problems for general gradient compressors is provided. By partitioning the gradient into blocks, a blockwise compressor is introduced such that each gradient block is compressed and transmitted in 1-bit format with a scaling factor, leading to a nearly 32x reduction on communication. Experimental results show that the proposed method converges as fast as full-precision distributed momentum SGD and achieves the same testing accuracy. In particular, on distributed ResNet training with 7 workers on the ImageNet, the proposed algorithm achieves the same testing accuracy as momentum SGD using full-precision gradients, but with $46\%$ less wall clock time.

## 1 Introduction

Deep neural networks have been highly successful in recent years [9, 10, 17, 22, 27]. To achieve state-of-the-art performance, they often have to leverage the computing power of multiple machines during training [8, 26, 28, 6]. Popular approaches include distributed synchronous SGD and its momentum variant SGDM, in which the computational load for evaluating a mini-batch gradient is distributed among the workers. Each worker performs local computation, and these local informations are then merged by the server for final update on the model parameters. However, its scalability is limited by the possibly overwhelming cost due to communication of the gradient and model parameter [12]. Let $d$ be the gradient/parameter dimensionality, and $M$ be the number of workers. $64Md$ bits need to be transferred between the workers and server in each iteration.

To mitigate this communication bottleneck, the two common approaches are gradient sparsification and gradient quantization. Gradient sparsification only sends the most significant, information-preserving gradient entries. A heuristic algorithm is first introduced in [16], in which only the large entries are transmitted. On training a neural machine translation model with 4 GPUs, this greatly reduces the communication overhead and achieves 22% speedup [1]. Deep gradient compression [13] is another heuristic method that combines gradient sparsification with other techniques such as momentum correction, local gradient clipping, and momentum factor masking, achieving significant

---

[*]The work was done before Shuai Zheng joined Amazon Web Services.

reduction on communication cost. Recently, a stochastic sparsification method was proposed in [23] that balances sparsity and variance by solving a constrained linear programming. MEM-SGD [18] combines top-$k$ sparsification with error correction. By keeping track of the accumulated errors, these can be added back to the gradient estimator before each transmission. MEM-SGD converges at the same rate as SGD on convex problems, whilst reducing the communication overhead by a factor equal to the problem dimensionality.

On the other hand, gradient quantization mitigates the communication bottleneck by lowering the gradient's floating-point precision with a smaller bit width. 1-bit SGD achieves state-of-the-art results on acoustic modeling while dramatically reducing the communication cost [16, 19]. TernGrad [24] quantizes the gradients to ternary levels $\{-1, 0, 1\}$. QSGD [2] employs stochastic randomized rounding to ensure unbiasedness of the estimator. Error-compensated quantized SGD (ECQ-SGD) was proposed in [25], wherein a similar stochastic quantization function used in QSGD is employed, and an error bound is obtained for quadratic loss functions. Different from the error-feedback mechanism proposed in MEM-SGD, ECQ-SGD requires two more hyper-parameters and its quantization errors are decayed exponentially. Thus, error feedback is limited to a small number of iterations. Also, ECQ-SGD uses all-to-all broadcast (which may involve large network traffic and idle time), while we consider parameter-server architecture. Recently, Bernstein *et al.* proposed signSGD with majority vote [3], which only transmits the 1-bit gradient sign between workers and server. A variant using momentum, called signum with majority vote, is also introduced though without convergence analysis [4] . Using the majority vote, signSGD achieves a notion of Byzantine fault tolerance [4]. Moreover, it converges at the same rate as distributed SGD, though it has to rely on the unrealistic assumptions of having a large mini-batch and unimodal symmetric gradient noise. Indeed, signSGD can diverge in some simple cases when these assumptions are violated [11]. With only a single worker, this divergence issue can be fixed by using the error correction technique in MEM-SGD, leading to SGD with error-feedback (EF-SGD) [11].

While only a single worker is considered in EF-SGD, we study in this paper the more interesting distributed setting. An extension of MEM-SGD and EF-SGD with parallel computing was proposed in [7] for all-to-all broadcast. Another related architecture is allreduce. Compression at the server can be implemented between the reduce and broadcast steps in tree allreduce, or between the reduce-scatter and allgather steps in ring allreduce. However, allreduce requires repeated gradient aggregations, and the compressed gradients need to be first decompressed before they are summed. Hence, heavy overheads may be incurred.

In this paper, we study the distributed setting with a parameter server architecture. To ensure efficient communication, we consider two-way gradient compression, in which gradients in both directions (server to/from workers) are compressed. Note that existing works (except signSGD/signum with majority vote [3, 4]) do not compress the aggregated gradients before sending back to workers. Moreover, as gradients in a deep network typically have similar magnitudes in each layer, each layer-wise gradient can be sufficiently represented using a sign vector and its average $\ell_1$-norm. This layer-wise (or blockwise in general) compressor achieves nearly 32x reduction in communication cost. The resulant procedure is called communication-efficient distributed SGD with error-feedback (dist-EF-SGD). Analogous to SGDM, we also propose a stochastic variant dist-EF-SGDM with Nesterov's momentum [14]. The convergence properties of dist-EF-SGD(M) are studied theoretically.

Our contributions are: (i) We provide a bound on dist-EF-SGD with general stepsize schedule for a class of compressors (including the commonly used sign-operator and top-$k$ sparsification). In particular, without relying on the unrealistic assumptions in [3, 4], we show that dist-EF-SGD with constant/decreasing/increasing stepsize converges at an $\mathcal{O}(1/\sqrt{MT})$ rate, which matches that of distributed synchronous SGD; (ii) We study gradient compression with Nesterov's momentum in a parameter server. For dist-EF-SGDM with constant stepsize, we obtain an $\mathcal{O}(1/\sqrt{MT})$ rate. To the best of our knowledge, these are the first convergence results on two-way gradient compression with Nesterov's momentum; (iii) We propose a general blockwise compressor and show its theoretical properties. Experimental results show that the proposed algorithms are efficient without losing prediction accuracy. After our paper has appeared, we note a similar idea was independently proposed in [21]. Different from ours, they do not consider changing stepsize, blockwise compressor and Nesterov's momentum.

**Notations**. For a vector $x$, $\|x\|_1$ and $\|x\|_2$ are its $\ell_1$- and $\ell_2$-norms, respectively. $\text{sign}(x)$ outputs a vector in which each element is the sign of the corresponding entry of $x$. For two vectors $x, y$, $\langle x, y \rangle$ denotes the dot product. For a function $f$, its gradient is $\nabla f$.

## 2  Related Work: SGD with Error-Feedback

In machine learning, one is often interested in minimizing the expected risk $F(x) = \mathbb{E}_\xi[f(x, \xi)]$. which directly measures the generalization error [5]. Here, $x \in \mathbb{R}^d$ is the model parameter, $\xi$ is drawn from some unknown distribution, and $f(x, \xi)$ is the possibly nonconvex risk due to $x$. When the expectation is taken over a training set of size $n$, the expected risk reduces to empirical risk.

Recently, Karimireddy *et al.* [11] introduced SGD with error-feedback (EF-SGD), which combines gradient compression with error correction (Algorithm 1). A single machine is considered, which keeps the gradient difference that is not used for parameter update in the current iteration. In the next iteration $t$, the accumulated residual $e_t$ is added to the current gradient. The corrected gradient $p_t$ is then fed into an $\delta$-approximate compressor.

**Definition 1.** *[11] An operator $\mathcal{C} : \mathbb{R}^d \rightarrow \mathbb{R}^d$ is an $\delta$-approximate compressor for $\delta \in (0, 1]$ if $\|\mathcal{C}(x) - x\|_2^2 \leq (1 - \delta)\|x\|_2^2$.*

Examples of $\delta$-approximate compressors include the scaled sign operator $\mathcal{C}(v) = \|v\|_1/d \cdot \text{sign}(v)$ [11] and top-$k$ operator (which only preserves the $k$ coordinates with the largest absolute values) [18]. One can also have randomized compressors that only satisfy Definition 1 in expectation. Obviously, it is desirable to have a large $\delta$ while achieving low communication cost.

---

**Algorithm 1** SGD with Error-Feedback (EF-SGD) [11]

1: **Input:** stepsize $\eta$; compressor $\mathcal{C}(\cdot)$.
2: **Initialize:** $x_0 \in \mathbb{R}^d$; $e_0 = 0 \in \mathbb{R}^d$
3: **for** $t = 0, \dots, T - 1$ **do**
4:     $p_t = \eta g_t + e_t$ {stochastic gradient $g_t = \nabla f(x_t, \xi_t)$}
5:     $\Delta_t = \mathcal{C}(p_t)$ {compressed value output}
6:     $x_{t+1} = x_t - \Delta_t$
7:     $e_{t+1} = p_t - \Delta_t$
8: **end for**

---

EF-SGD achieves the same $\mathcal{O}(1/\sqrt{T})$) rate as SGD. To obtain this convergence guarantee, an important observation is that the error-corrected iterate $\tilde{x}_t = x_t - e_t$ satisfies the recurrence: $\tilde{x}_{t+1} = \tilde{x}_t - \eta g_t$, which is similar to that of SGD. This allows utilizing the convergence proof of SGD to bound the gradient difference $\|\nabla F(\tilde{x}_t) - \nabla F(x_t)\|_2$.

## 3  Distributed Blockwise Momentum SGD with Error-Feedback

### 3.1  Distributed SGD with Error-Feedback

The proposed procedure, which extends EF-SGD to the distributed setting. is shown in Algorithm 2. The computational workload is distributed over $M$ workers. A local accumulated error vector $e_{t,i}$ and a local corrected gradient vector $p_{t,i}$ are stored in the memory of worker $i$. At iteration $t$, worker $i$ pushes the compressed signal $\Delta_{t,i} = \mathcal{C}(p_{t,i})$ to the parameter server. On the server side, all workers' $\Delta_{t,i}$'s are aggregated and used to update its global error-corrected vector $\tilde{p}_t$. Before sending back the final update direction $\tilde{p}_t$ to each worker, compression is performed to ensure a comparable amount of communication costs between the push and pull operations. Due to gradient compression on the server, we also employ a global accumulated error vector $\tilde{e}_t$. Unlike EF-SGD in Algorithm 1, we do not multiply gradient $g_{t,i}$ by the stepsize $\eta_t$ before compression. The two cases make no difference when $\eta_t$ is constant. However, when the stepsize is changing over time, this would affect convergence. We also rescale the local accumulated error $e_{t,i}$ by $\eta_{t-1}/\eta_t$. This modification, together with the use of error correction on both workers and server, allows us to obtain Lemma 1. Because of these differences, note that dist-EF-SGD does not reduce to EF-SGD when $M = 1$. When $\mathcal{C}(\cdot)$ is the identity mapping, dist-EF-SGD reduces to full-precision distributed SGD.

---

**Algorithm 2** Distributed SGD with Error-Feedback (dist-EF-SGD)

---

1: **Input:** stepsize sequence $\{\eta_t\}$ with $\eta_{-1} = 0$; number of workers $M$; compressor $\mathcal{C}(\cdot)$.
2: **Initialize:** $x_0 \in \mathbb{R}^d$; $e_{0,i} = 0 \in \mathbb{R}^d$ on each worker $i$; $\tilde{e}_0 = 0 \in \mathbb{R}^d$ on server
3: **for** $t = 0, \ldots, T-1$ **do**
4:     **on each worker** $i$
5:         $p_{t,i} = g_{t,i} + \frac{\eta_{t-1}}{\eta_t} e_{t,i}$ {stochastic gradient $g_{t,i} = \nabla f(x_t, \xi_{t,i})$}
6:         **push** $\Delta_{t,i} = \mathcal{C}(p_{t,i})$ **to server**
7:         $x_{t+1} = x_t - \eta_t \tilde{\Delta}_t$ {$\tilde{\Delta}_t$ **is pulled from server**}
8:         $e_{t+1,i} = p_{t,i} - \Delta_{t,i}$
9:     **on server**
10:         **pull** $\Delta_{t,i}$ **from each worker** $i$ and $\tilde{p}_t = \frac{1}{M} \sum_{i=1}^{M} \Delta_{t,i} + \frac{\eta_{t-1}}{\eta_t} \tilde{e}_t$
11:         **push** $\tilde{\Delta}_t = \mathcal{C}(\tilde{p}_t)$ **to each worker**
12:         $\tilde{e}_{t+1} = \tilde{p}_t - \tilde{\Delta}_t$
13: **end for**

---

In the following, we investigate the convergence of dist-EF-SGD. We make the following assumptions, which are common in the stochastic approximation literature.

**Assumption 1.** *F is lower-bounded (i.e., $F_* = \inf_{x \in \mathbb{R}^d} F(x) > -\infty$) and L-smooth (i.e., $F(x) \leq F(y) + \langle \nabla F(y), x - y \rangle + \frac{L}{2} \|x - y\|_2^2$ for $x, y \in \mathbb{R}^d$).*

**Assumption 2.** *The stochastic gradient $g_{t,i}$ has bounded variance: $\mathbb{E}_t \left[ \|g_{t,i} - \nabla F(x_t)\|_2^2 \right] \leq \sigma^2$.*

**Assumption 3.** *The full gradient $\nabla F$ is uniformly bounded: $\|\nabla F(x_t)\|_2^2 \leq \omega^2$.*

This implies the second moment is bounded, i.e., $\mathbb{E}_t \left[ \|g_{t,i}\|_2^2 \right] \leq G^2 \equiv \sigma^2 + \omega^2$.

**Lemma 1.** *Consider the error-corrected iterate $\tilde{x}_t = x_t - \eta_{t-1} \left( \tilde{e}_t + \frac{1}{M} \sum_{i=1}^{M} e_{t,i} \right)$, where $x_t$, $\tilde{e}_t$, and $e_{t,i}$'s are generated from Algorithm 2. It satisfies the recurrence: $\tilde{x}_{t+1} = \tilde{x}_t - \eta_t \frac{1}{M} \sum_{i=1}^{M} g_{t,i}$.*

The above Lemma shows that $\tilde{x}_t$ is very similar to the distributed SGD iterate except that the stochastic gradients are evaluated at $x_t$ instead of $\tilde{x}_t$. This connection allows us to utilize the analysis of full-precision distributed SGD. In particular, we have the following Lemma.

**Lemma 2.** $\mathbb{E} \left[ \left\| \tilde{e}_t + \frac{1}{M} \sum_{i=1}^{M} e_{t,i} \right\|_2^2 \right] \leq \frac{8(1-\delta)G^2}{\delta^2} \left[ 1 + \frac{16}{\delta^2} \right]$ *for any $t \geq 0$.*

This implies that $\nabla F(\tilde{x}_t) \approx \nabla F(x_t)$ by Assumption 1. Given the above results, we can prove convergence of the proposed method by utilizing tools used on the full-precision distributed SGD.

**Theorem 1.** *Suppose that Assumptions 1-3 hold. Assume that $0 < \eta_t < 3/(2L)$ for all t. For the $\{x_t\}$ sequence generated from Algorithm 2, we have*

$$
\mathbb{E} \left[ \|\nabla F(x_o)\|_2^2 \right] \leq \frac{4}{\sum_{k=0}^{T-1} \eta_k (3 - 2L\eta_k)} [F(x_0) - F_*] + \frac{2L\sigma^2}{M} \sum_{t=0}^{T-1} \frac{\eta_t^2}{\sum_{k=0}^{T-1} \eta_k (3 - 2L\eta_k)}
$$

$$
+ \frac{32L^2(1-\delta)G^2}{\delta^2} \left[ 1 + \frac{16}{\delta^2} \right] \sum_{t=0}^{T-1} \frac{\eta_t \eta_{t-1}^2}{\sum_{k=0}^{T-1} \eta_k (3 - 2L\eta_k)},
$$

*where $o \in \{0, \ldots, T-1\}$ is an index such that $P(o = k) = \frac{\eta_k (3 - 2L\eta_k)}{\sum_{t=0}^{T-1} \eta_t (3 - 2L\eta_t)}$, $\forall k = 0, \ldots, T-1$.*

The first term on the RHS shows decay of the initial value. The second term is related to the variance, and the proposed algorithm enjoys variance reduction with more workers. The last term is due to gradient compression. A large $\delta$ (less compression) makes this term smaller and thus faster convergence. Similar to the results in [11], our bound also holds for unbiased compressors (e.g., QSGD [2]) of the form $\mathcal{C}(\cdot) = cU(\cdot)$, where $\mathbb{E}[U(x)] = x$ and $\mathbb{E}[\|U(x)\|_2^2] \leq \frac{1}{c} \|x\|_2^2$ for some $0 < c < 1$. Then, $cU(\cdot)$ is a $c$-approximate compressor in expectation.

The following Corollary shows that dist-EF-SGD has a convergence rate of $\mathcal{O}(1/\sqrt{MT})$, leading to a $\mathcal{O}(1/(M\epsilon^4))$ iteration complexity for satisfying $\mathbb{E}[\|\nabla F(x_o)\|_2^2] \leq \epsilon^2$.

---

**Algorithm 3** Distributed Blockwise SGD with Error-Feedback (dist-EF-blockSGD)

---

1: **Input:** stepsize sequence $\{\eta_t\}$ with $\eta_{-1} = 0$; number of workers $M$; block partition $\{\mathcal{G}_1, \ldots, \mathcal{G}_B\}$.
2: **Initialize:** $x_0 \in \mathbb{R}^d$; $e_{0,i} = 0 \in \mathbb{R}^d$ on each worker $i$; $\tilde{e}_0 = 0 \in \mathbb{R}^d$ on server
3: **for** $t = 0, \ldots, T-1$ **do**
4:     **on each worker** $i$
5:         $p_{t,i} = g_{t,i} + \frac{\eta_{t-1}}{\eta_t} e_{t,i}$ {stochastic gradient $g_{t,i} = \nabla f(x_t, \xi_{t,i})$}
6:         **push** $\Delta_{t,i} = \left[ \frac{\|p_{t,i,\mathcal{G}_1}\|_1}{d_1} \text{sign}(p_{t,i,\mathcal{G}_1}), \ldots, \frac{\|p_{t,i,\mathcal{G}_B}\|_1}{d_B} \text{sign}(p_{t,i,\mathcal{G}_B}) \right]$ **to server**
7:         $x_{t+1} = x_t - \eta_t \tilde{\Delta}_t$ {$\tilde{\Delta}_t$ **is pulled from server**}
8:         $e_{t+1,i} = p_{t,i} - \Delta_{t,i}$
9:     **on server**
10:        **pull** $\Delta_{t,i}$ **from each worker** $i$ and $\tilde{p}_t = \frac{1}{M} \sum_{i=1}^{M} \Delta_{t,i} + \frac{\eta_{t-1}}{\eta_t} \tilde{e}_t$
11:        **push** $\tilde{\Delta}_t = \left[ \frac{\|\tilde{p}_{t,\mathcal{G}_1}\|_1}{d_1} \text{sign}(\tilde{p}_{t,\mathcal{G}_1}), \ldots, \frac{\|\tilde{p}_{t,\mathcal{G}_B}\|_1}{d_B} \text{sign}(\tilde{p}_{t,\mathcal{G}_B}) \right]$ **to each worker**
12:        $\tilde{e}_{t+1} = \tilde{p}_t - \tilde{\Delta}_t$
13: **end for**

---

**Corollary 1.** *Let stepsize* $\eta = \min(\frac{1}{2L}, \frac{\gamma}{\sqrt{T}/\sqrt{M} + (1-\delta)^{1/3}(1/\delta^2 + 16/\delta^4)^{1/3} T^{1/3}})$ *for some* $\gamma > 0$. *Then,*

$$
\begin{aligned}
\mathbb{E}[\|\nabla F(x_o)\|_2^2] \leq \quad &\frac{4L}{T}[F(x_0) - F_*] + \left[ \frac{2}{\gamma}[F(x_0) - F_*] + L\gamma\sigma^2 \right] \frac{1}{\sqrt{MT}} \\
&+ \frac{2(1-\delta)^{1/3} \left[ \frac{1}{\gamma}[F(x_0) - F_*] + 8L^2\gamma^2 G^2 \right]}{\delta^{2/3} T^{2/3}} \left[ 1 + \frac{16}{\delta^2} \right]^{1/3}.
\end{aligned}
$$

*In comparison, under the same assumptions, distributed synchronous SGD achieves*

$$
\mathbb{E}[\|\nabla F(x_o)\|_2^2] \leq \frac{8L}{3T}[F(x_0) - F_*] + \left[ \frac{2}{\gamma}[F(x_0) - F_*] + L\gamma\sigma^2 \right] \frac{2}{3\sqrt{MT}}.
$$

Thus, the convergence rate of dist-EF-SGD matches that of distributed synchronous SGD (with full-precision gradients) after $T \geq O(1/\delta^2)$ iterations, even though gradient compression is used. Moreover, more workers (larger $M$) leads to faster convergence. Note that the bound above does not reduce to that of EF-SGD when $M = 1$, as we have two-way compression. When $M = 1$, our bound also differs from Remark 4 in [11] in that our last term is $O((1-\delta)^{1/3}/(\delta^{4/3}T^{2/3}))$, while theirs is $O((1-\delta)/(\delta^2 T))$ (which is for single machine with one-way compression). Ours is worse by a factor of $O(T^{1/3}\delta^{2/3}/(1-\delta)^{2/3})$, which is the price to pay for two-way compression and a linear speedup of using $M$ workers. Moreover, unlike signSGD with majority vote [3], we achieve a convergence rate of $\mathcal{O}(1/\sqrt{MT})$ without assuming a large mini-batch size ($= T$) and unimodal symmetric gradient noise.

Theorem 1 only requires $0 < \eta_t < 3/(2L)$ for all $t$. This thus allows the use of any decreasing, increasing, or hybrid stepsize schedule. In particular, we have the following Corollary.

**Corollary 2.** *Let* $\eta_t = \frac{\gamma}{((t+1)T)^{1/4}/(\sqrt{M}) + (1-\delta)^{1/3}(1/\delta^2 + 16/\delta^4)^{1/3} T^{1/3}}$ *(decreasing stepsize) with* $T \geq 16L^4\gamma^4 M^2$ *or* $\eta_t = \frac{\gamma\sqrt{t+1}}{T/\sqrt{M} + (1-\delta)^{1/3}(1/\delta^2 + 16/\delta^4)^{1/3} T^{5/6}}$ *(increasing stepsize) with* $T \geq 4L^2\gamma^2 M$. *Then, dist-EF-SGD converges to a stationary point at a rate of* $\mathcal{O}(1/\sqrt{MT})$.

To the best of our knowledge, this is the first such result for distributed compressed SGD with decreasing/increasing stepsize on nonconvex problems. These two stepsize schedules can also be used together. For example, one can use an increasing stepsize at the beginning of training as warm-up, and then a decreasing stepsize afterwards.

## 3.2 Blockwise Compressor

A commonly used compressor is [11]:

$$
\mathcal{C}(v) = \|v\|_1/d \cdot \text{sign}(v). \tag{1}
$$

Compared to using only the sign operator as in signSGD, the factor $\|v\|_1/d$ can preserve the gradient's magnitude. However, as shown in [11], its $\delta$ in Definition 1 is $\|v\|_1^2/(d\|v\|_2^2)$, and can be particularly small when $v$ is sparse. When $\delta$ is closer to 1, the bound in Corollary 1 becomes smaller and thus convergence is faster. In this section, we achieve this by proposing a blockwise extension of (1).

Specifically, we partition the compressor input $v$ into $B$ blocks, where each block $b$ has $d_b$ elements indexed by $\mathcal{G}_b$. Block $b$ is then compressed with scaling factor $\|v_{\mathcal{G}_b}\|_1/d_b$ (where $v_{\mathcal{G}_b}$ is the subvector of $v$ with elements in block $b$), leading to: $\mathcal{C}_B(v) = [\|v_{\mathcal{G}_1}\|_1/d_1 \cdot \text{sign}(v_{\mathcal{G}_1}),\ldots,\|v_{\mathcal{G}_B}\|_1/d_B \cdot \text{sign}(v_{\mathcal{G}_B})]$. A similar compression scheme, with each layer being a block, is considered in the experiments of [11]. However, they provide no theoretical justifications. The following Proposition first shows that $\mathcal{C}_B(\cdot)$ is also an approximate compressor.

**Proposition 1.** *Let* $[B] = \{1,2,\ldots,B\}$. $\mathcal{C}_B$ *is a* $\phi(v)$-*approximate compressor, where* $\phi(v) = \min_{b\in[B]} \frac{\|v_{\mathcal{G}_b}\|_1^2}{d_b\|v_{\mathcal{G}_b}\|_2^2} \geq \min_{b\in[B]} \frac{1}{d_b}$.

The resultant algorithm will be called dist-EF-blockSGD (Algorithm 3) in the sequel. As can be seen, this is a special case of Algorithm 2. By replacing $\delta$ with $\phi(v)$ in Proposition 1, the convergence results of dist-EF-SGD in Section 3.1 can be directly applied.

There are many ways to partition the gradient into blocks. In practice, one can simply consider each parameter tensor/matrix/vector in the deep network as a block. The intuition is that (i) gradients in the same parameter tensor/matrix/vector typically have similar magnitudes, and (ii) the corresponding scaling factors can thus be tighter than the scaling factor obtained on the whole parameter, leading to a larger $\delta$. As an illustration of (i), Figure 1(a) shows the coefficient of variation (which is defined as the ratio of the standard deviation to the mean) of $\{|g_{t,i}|\}_{i\in\mathcal{G}_b}$ averaged over all blocks and iterations in an epoch, obtained from ResNet-20 on the CIFAR-100 dataset (with a mini-batch size of 16 per worker).[2] A value smaller than 1 indicates that the absolute gradient values in each block concentrate around the mean. As for point (ii) above, consider the case where all the blocks are of the same size ($d_b = \tilde{d}, \forall b$), elements in the same block have the same magnitude ($\forall i \in \mathcal{G}_b, |v_i| = c_b$ for some $c_b$), and the magnitude is increasing across blocks ($c_b/c_{b+1} = \alpha$ for some $\alpha < 1$). For the standard compressor in (1), $\delta = \frac{\|v\|_1^2}{d\|v\|_2^2} = \frac{(1+\alpha)(1-\alpha^B)}{B(1-\alpha)(1+\alpha^B)} \approx \frac{(1+\alpha)}{B(1-\alpha)}$ for a sufficiently large $B$; whereas for the proposed blockwise compressor, $\phi(v) = 1 \gg \frac{(1+\alpha)}{B(1-\alpha)}$. Figure 1(b) shows the empirical estimates of $\|v\|_1^2/(d\|v\|_2^2)$ and $\phi(v)$ in the ResNet-20 experiment. As can be seen, $\phi(v) \gg \|v\|_1^2/(d\|v\|_2^2)$.

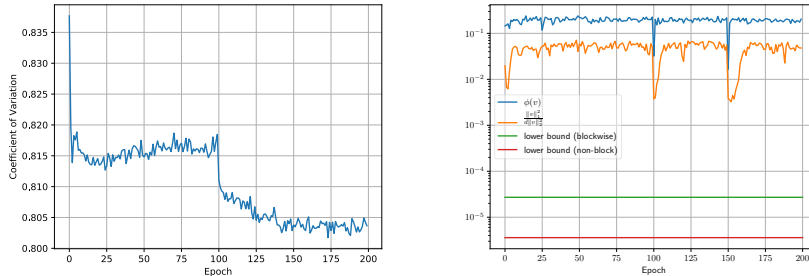

(a) Coefficient of variation of $\{|g_{t,i}|\}_{i\in\mathcal{G}_b}$.    (b) $\delta$ for blockwise and non-block versions.

Figure 1: Illustrations using the ResNet-20 in Section 4.1. Left: Averaged coefficient of variation of $\{|g_{t,i}|\}_{i\in\mathcal{G}_b}$. Right: Empirical estimates of $\delta$ for the blockwise ($\phi(v)$ in Proposition 1) and non-block versions ($\|v\|_1^2/(d\|v\|_2^2)$). Each point is the minimum among all iterations in an epoch. The lower bounds, $\min_{b\in[B]} 1/d_b$ and $1/d$, are also shown. Note that the ordinate is in log scale.

The per-iteration communication costs of the various distributed algorithms are shown in Table 1. Compared to signSGD with majority vote [3], dist-EF-blockSGD requires an extra $64MB$ bits for transmitting the blockwise scaling factors (each factor $\|v_{\mathcal{G}_b}\|_1/d_b$ is stored in float32 format and transmitted twice in each iteration). By treating each vector/matrix/tensor parameter as a block, $B$ is typically in the order of hundreds. For most problems of interest, $64MB/(2Md) < 10^{-3}$. The reduction in communication cost compared to full-precision distributed SGD is thus nearly 32x.

Table 1: Communication costs of the various distributed gradient compression algorithms and SGD.

| algorithm | #bits per iteration |
|---|---|
| full-precision SGD | $64Md$ |
| signSGD with majority vote | $2Md$ |
| dist-EF-blockSGD | $2Md + 64MB$ |

### 3.3 Nesterov's Momentum

Momentum has been widely used in deep networks [20]. Standard distributed SGD with Nesterov's momentum [14] and full-precision gradients uses the update: $m_{t,i} = \mu m_{t-1,i} + g_{t,i}, \forall i \in [M]$ and $x_{t+1} = x_t - \eta_t \frac{1}{M} \sum_{i=1}^{M} (\mu m_{t,i} + g_{t,i})$, where $m_{t,i}$ is a local momentum vector maintained by each worker $i$ at time $t$ (with $m_{0,i} = 0$), and $\mu \in [0, 1)$ is the momentum parameter. In this section, we extend the proposed dist-EF-SGD with momentum. Instead of sending the compressed $g_{t,i} + \frac{\eta_{t-1}}{\eta_t} e_{t,i}$ to the server, the compressed $\mu m_{t,i} + g_{t,i} + \frac{\eta_{t-1}}{\eta_t} e_{t,i}$ is sent. The server merges all the workers's results and sends it back to each worker. The resultant procedure with blockwise compressor is called dist-EF-blockSGDM (Algorithm 4), and has the same communication cost as dist-EF-blockSGD. The corresponding non-block variant is analogous.

---

**Algorithm 4** Distributed Blockwise Momentum SGD with Error-Feedback (dist-EF-blockSGDM)

1: **Input:** stepsize sequence $\{\eta_t\}$ with $\eta_{-1} = 0$; momentum parameter $0 \le \mu < 1$; number of workers $M$; block partition $\{\mathcal{G}_1, \ldots, \mathcal{G}_B\}$.
2: **Initialize:** $x_0 \in \mathbb{R}^d$; $m_{-1,i} = e_{0,i} = 0 \in \mathbb{R}^d$ on each worker $i$; $\tilde{e}_0 = 0 \in \mathbb{R}^d$ on server
3: **for** $t = 0, \ldots, T - 1$ **do**
4:     **on each worker** $i$
5:         $m_{t,i} = \mu m_{t-1,i} + g_{t,i}$ {stochastic gradient $g_{t,i} = \nabla f(x_t, \xi_{t,i})$}
6:         $p_{t,i} = \mu m_{t,i} + g_{t,i} + \frac{\eta_{t-1}}{\eta_t} e_{t,i}$
7:         **push** $\Delta_{t,i} = \left[ \frac{\|p_{t,i,\mathcal{G}_1}\|_1}{d_1} \text{sign}(p_{t,i,\mathcal{G}_1}), \ldots, \frac{\|p_{t,i,\mathcal{G}_B}\|_1}{d_B} \text{sign}(p_{t,i,\mathcal{G}_B}) \right]$ **to server**
8:         $x_{t+1} = x_t - \eta_t \tilde{\Delta}_t$ {$\tilde{\Delta}_t$ **is pulled from server**}
9:         $e_{t+1,i} = p_{t,i} - \Delta_{t,i}$
10:     **on server**
11:         **pull** $\Delta_{t,i}$ **from each worker** $i$ and $\tilde{p}_t = \frac{1}{M} \sum_{i=1}^{M} \Delta_{t,i} + \frac{\eta_{t-1}}{\eta_t} \tilde{e}_t$
12:         **push** $\tilde{\Delta}_t = \left[ \frac{\|\tilde{p}_{t,\mathcal{G}_1}\|_1}{d_1} \text{sign}(\tilde{p}_{t,\mathcal{G}_1}), \ldots, \frac{\|\tilde{p}_{t,\mathcal{G}_B}\|_1}{d_B} \text{sign}(\tilde{p}_{t,\mathcal{G}_B}) \right]$ **to each worker**
13:         $\tilde{e}_{t+1} = \tilde{p}_t - \tilde{\Delta}_t$
14: **end for**

---

Similar to Lemma 1, the following Lemma shows that the error-corrected iterate $\tilde{x}_t$ is very similar to Nesterov's accelerated gradient iterate, except that the momentum is computed based on $\{x_t\}$.

**Lemma 3.** *The error-corrected iterate $\tilde{x}_t = x_t - \eta_{t-1}(\tilde{e}_t + \frac{1}{M} \sum_{i=1}^{M} e_{t,i})$, where $x_t$, $\tilde{e}_t$, and $e_{t,i}$'s are generated from Algorithm 4, satisfies the recurrence: $\tilde{x}_{t+1} = \tilde{x}_t - \eta_t \frac{1}{M} \sum_{i=1}^{M} (\mu m_{t,i} + g_{t,i})$.*

As in Section 3.1, it can be shown that $\|\tilde{e}_t + \frac{1}{M} \sum_{i=1}^{M} e_{t,i}\|_2$ is bounded and $\nabla F(\tilde{x}_t) \approx \nabla F(x_t)$. The following Theorem shows the convergence rate of the proposed dist-EF-blockSGDM.

**Theorem 2.** *Suppose that Assumptions 1-3 hold. Let $\eta_t = \eta$ for some $\eta > 0$. For any $\eta \le \frac{(1-\mu)^2}{2L}$, and the $\{x_t\}$ sequence generated from Algorithm 4, we have*

$$\mathbb{E}\left[ \|\nabla F(x_o)\|_2^2 \right] \le \frac{4(1-\mu)}{\eta T}[F(x_0) - F_*] + \frac{2L\eta\sigma^2}{(1-\mu)M}\left[ 1 + \frac{2L\eta\mu^4}{(1-\mu)^3} \right] \qquad (2)$$
$$+ \frac{32L^2\eta^2(1-\delta)G^2}{\delta^2(1-\mu)^2}\left[ 1 + \frac{16}{\delta^2} \right].$$

Compared to Theorem 1, using a larger momentum parameter $\mu$ makes the first term (which depends on the initial condition) smaller but a worse variance term (second term) and error term due to gradient compression (last term). Similar to Theorem 1, a larger $\eta$ makes the third term larger. The following Corollary shows that the proposed dist-EF-blockSGDM achieves a convergence rate of $\mathcal{O}(((1-\mu)[F(x_0) - F_*] + \sigma^2/(1-\mu))/\sqrt{MT})$.

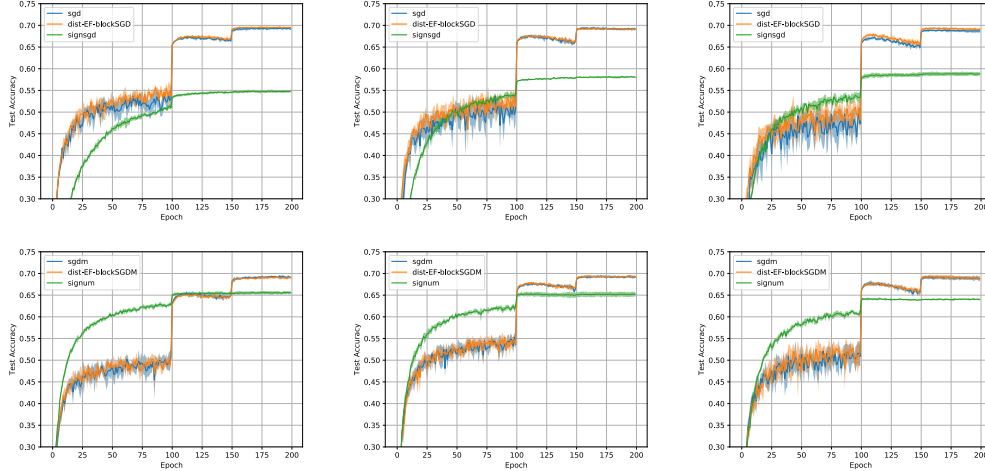

(a) Mini-batch size: 8 per worker. (b) Mini-batch size: 16 per worker. (c) Mini-batch size: 32 per worker.

Figure 2: Testing accuracy on CIFAR-100. Top: No momentum; Bottom: With momentum. The solid curve is the mean accuracy over five repetitions. The shaded region spans one standard deviation.

**Corollary 3.** *Let* $\eta = \frac{\gamma}{\sqrt{T}/\sqrt{M}+(1-\delta)^{1/3}(1/\delta^2+16/\delta^4)^{1/3}T^{1/3}}$ *for some* $\gamma > 0$. *For any* $T \geq \frac{4\gamma^2 L^2 M}{(1-\mu)^4}$, $\mathbb{E}\left[\|\nabla F(x_o)\|_2^2\right] \leq \left[\frac{2(1-\mu)}{\gamma}[F(x_0) - F_*] + \frac{L\gamma\sigma^2}{1-\mu}\right]\frac{2}{\sqrt{MT}} + \frac{4L^2\gamma^2\mu^4\sigma^2}{(1-\mu)^4 T} + \frac{4(1-\delta)^{1/3}\left[\frac{(1-\mu)}{\gamma}[F(x_0)-F_*]+8L^2\gamma^2 G^2/(1-\mu)^2\right]}{\delta^{2/3}T^{2/3}}\left[1 + \frac{16}{\delta^2}\right]^{1/3}.$

# 4 Experiments

## 4.1 Multi-GPU Experiment on CIFAR-100

In this experiment, we demonstrate that the proposed dist-EF-blockSGDM and dist-EF-blockSGD ($\mu = 0$ in Algorithm 4), though using fewer bits for gradient transmission, still has good convergence. For faster experimentation, we use a single node with multiple GPUs (an AWS P3.16 instance with 8 Nvidia V100 GPUs, each GPU being a worker) instead of a distributed setting.

Experiment is performed on the CIFAR-100 dataset, with 50K training images and 10K test images. We use a 20-layer ResNet [10]. Each parameter tensor/matrix/vector is treated as a block in dist-EF-blockSGD(M). They are compared with (i) distributed synchronous SGD (with full-precision gradient); (ii) distributed synchronous SGD (full-precision gradient) with momentum (SGDM); (iii) signSGD with majority vote [3]; and (iv) signum with majority vote [4]. All the algorithms are implemented in MXNet. We vary the mini-batch size per worker in $\{8, 16, 32\}$. Results are averaged over 5 repetitions. More details of the experiments are shown in Appendix A.1.

Figure 2 shows convergence of the testing accuracy w.r.t. the number of epochs. As can be seen, dist-EF-blockSGD converges as fast as SGD and has slightly better accuracy, while signSGD performs poorly. In particular, dist-EF-blockSGD is robust to the mini-batch size, while the performance of signSGD degrades with smaller mini-batch size (which agrees with the results in [3]). Momentum makes SGD and dist-EF-blockSGD faster with mini-batch size of 16 or 32 per worker, particularly before epoch 100. At epoch 100, the learning rate is reduced, and the difference is less obvious. This is because a larger mini-batch means smaller variance $\sigma^2$, so the initial optimality gap $F(x_0) - F_*$ in (2) is more dominant. Use of momentum ($\mu > 0$) is then beneficial. On the other hand, momentum significantly improves signSGD. However, signum is still much worse than dist-EF-blockSGDM.

## 4.2 Distributed Training on ImageNet

In this section, we perform distributed optimization on ImageNet [15] using a 50-layer ResNet. Each worker is an AWS P3.2 instance with 1 GPU, and the parameter server is housed in one node. We

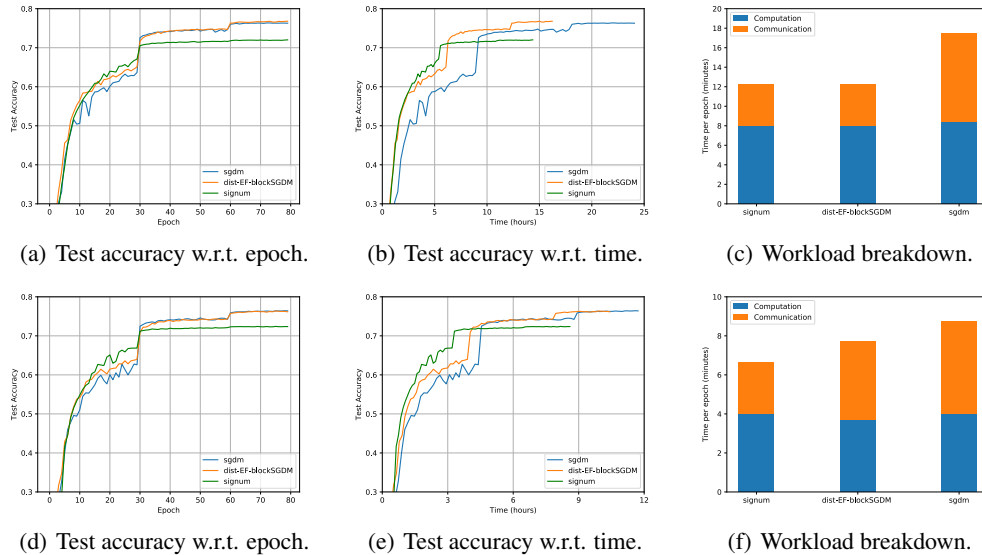

(a) Test accuracy w.r.t. epoch.    (b) Test accuracy w.r.t. time.    (c) Workload breakdown.

(d) Test accuracy w.r.t. epoch.    (e) Test accuracy w.r.t. time.    (f) Workload breakdown.

Figure 3: Distributed training results on the ImageNet dataset. Top: 7 workers; Bottom: 15 workers.

use the publicly available code[3] in [4], and the default communication library Gloo communication library in PyTorch. As in [4], we use its allreduce implementation for SGDM, which is faster.

As momentum accelerates the training for large mini-batch size in Section 4.1, we only compare the momentum variants here. The proposed dist-EF-blockSGDM is compared with (i) distributed synchronous SGD with momentum (SGDM); and (ii) signum with majority vote [4]. The number of workers $M$ is varied in $\{7, 15\}$. With an odd number of workers, a majority vote will not produce zero, and so signum does not lose accuracy by using 1-bit compression. More details of the setup are in Appendix A.2.

Figure 3 shows the testing accuracy w.r.t. the number of epochs and wall clock time. As in Section 4.1, SGDM and dist-EF-blockSGDM have comparable accuracies, while signum is inferior. When 7 workers are used, dist-EF-blockSGDM has higher accuracy than SGDM (76.77% vs 76.27%). dist-EF-blockSGDM reaches SGDM's highest accuracy in around 13 hours, while SGDM takes 24 hours (Figure 3(b)), leading to a 46% speedup. With 15 machines, the improvement is smaller (Figure 3(e)). This is because the burden on the parameter server is heavier. We expect comparable speedup with the 7-worker setting can be obtained by using more parameter servers. In both cases, signum converges fast but the test accuracies are about 4% worse.

Figures 3(c) and 3(f) show a breakdown of wall clock time into computation and communication time.[4] All methods have comparable computation costs, but signum and dist-EF-blockSGDM have lower communication costs than SGDM. The communication costs for signum and dist-EF-blockSGDM are comparable for 7 workers, but for 15 workers signum is lower. We speculate that it is because the sign vectors and scaling factors are sent separately to the server in our implementation, which causes more latency on the server with more workers. This may be alleviated if the two operations are fused.

## 5 Conclusion

In this paper, we proposed a distributed blockwise SGD algorithm with error feedback and momentum. By partitioning the gradients into blocks, we can transmit each block of gradient using 1-bit quantization with its average $\ell_1$-norm. The proposed methods are communication-efficient and have the same convergence rates as full-precision distributed SGD/SGDM for nonconvex objectives. Experimental results show that the proposed methods have fast convergence and achieve the same test accuracy as SGD/SGDM, while signSGD and signum only achieve much worse accuracies.

## Footnotes

[2]The detailed experimental setup is in Section 4.1.

[3]https://github.com/PermiJW/signSGD-with-Majority-Vote

[4]Following [4], communication time includes the extra computation time for error feedback and compression.

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
