[Supplementary Material]

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

# A    Experimental Setup

As we focus on synchronous distributed training, it is not necessary to compress weight decay. In the experiment, for dist-EF-blockSGD, the weight decay is not added to $g_{t,i}$. Instead, we add it to $\tilde{\Delta}_t$. For dist-EF-blockSGDM, as momentum is additive, we maintain an extra momentum $\tilde{m}_t$ for weight decay on each machine. Specifically, we perform the following update on each worker:

$$
\begin{aligned}
\tilde{m}_t &= \mu\tilde{m}_{t-1} + \lambda x_t, \\
x_{t+1} &= x_t - \eta_t(\tilde{\Delta}_t + \mu\tilde{m}_t + \lambda x_t),
\end{aligned}
$$

where $\lambda$ is the weight decay parameter. In the experiment, the sign is mapped to $\{-1, 1\}$ and takes 1 bit. Note that the gradient sign has zero probability of being zero.

## A.1    Setup: Multi-GPU Experiment on CIFAR-100

Each algorithm is run for 200 epochs. We only tune the initial stepsize, using a validation set with 5K images that is carved out from the training set. For dist-EF-blockSGD (resp. dist-EF-blockSGDM), we use the stepsize tuned for SGD (resp. SGDM). The stepsize with the best validation set performance is used to run the algorithm on the full training set. The stepsize is divided by 10 at the 100-th and 150-th epochs. The weight decay parameter is fixed to 0.0005, and the momentum parameter $\mu$ is 0.9. When mini-batch size is 16 per worker, for both SGD and SGDM, the stepsize is tuned from $\{0.05, 0.1, 0.5, 1\}$, and for signSGD and signum, the stepsize is chosen from $\{0.0005, 0.001, 0.005, 0.01\}$. When we obtain the best stepsize $\eta_0$ tuned with mini-batch size $B = 16$ per worker, for $B = 8$, the best stepsize is selected from $\{\eta_0/2, \eta_0\}$; whereas for $B = 32$, it is selected from $\{\eta_0, 2\eta_0\}$. The best stepsizes obtained are shown in Table 2

Table 2: Best stepsizes obtained by grid search on a hold-out validation set. We reuse the obtained stepsizes tuned for SGD/SGDM for dist-EF-blockSGD/dist-EF-blockSGDM.

|  | mini-batch size per worker | | |
| --- | --- | --- | --- |
| algorithm | 8 | 16 | 32 |
| full-precision SGD | 0.25 | 0.5 | 1 |
| full-precision SGDM | 0.05 | 0.05 | 0.1 |
| dist-EF-blockSGD | 0.25 | 0.5 | 1 |
| dist-EF-blockSGDM | 0.05 | 0.05 | 0.1 |
| signSGD | 0.001 | 0.001 | 0.002 |
| signum | 0.0005 | 0.0005 | 0.0005 |

## A.2    Setup: Distributed Training on ImageNet

We use the default hyperparameters for SGDM and signum in the code base, which have been tuned for the ImageNet experiment in [4]. Specifically, the momentum parameter $\mu$ is 0.9, and weight decay parameter is 0.0001. A mini-batch size of 128 per worker is employed.

For SGDM, we use $\eta = 0.1M$ (used for SGDM on the ImageNet experiment in the code base). For signum, $\eta = 0.0001$ (used for signum on the ImageNet experiment in the code base) on 7 workers and $\eta = 0.0002$ on 15 workers. For dist-EF-blockSGDM, we also use $\mu = 0.9$ and a weight decay of 0.0001. Its stepsize $\eta$ is 0.1 for 7 workers,[5] and 0.2 for 15 workers.

# B  Proof of Lemmas 1 and 3

**Lemma 4.** *Suppose that $p_{t,i} = z_{t,i} + \frac{\eta_{t-1}}{\eta_t} e_{t,i}$ for any sequence $z_{t,i}$. Consider the error-corrected iterate $\tilde{x}_t = x_t - \eta_{t-1} \left( \tilde{e}_t + \frac{1}{M} \sum_{i=1}^{M} e_{t,i} \right)$, it satisfies the recurrence:*

$$\tilde{x}_{t+1} = \tilde{x}_t - \eta_t \frac{1}{M} \sum_{i=1}^{M} z_{t,i}.$$

*Proof.*

$$\begin{aligned}
\tilde{x}_{t+1} &= x_t - \eta_t \mathcal{C}(\tilde{p}_t) - \eta_t \tilde{e}_{t+1} - \eta_t \frac{1}{M} \sum_{i=1}^{M} e_{t+1,i} && \text{Apply } x_{t+1} = x_t - \eta_t \mathcal{C}(\tilde{p}_t) \\
&= x_t - \eta_t \tilde{p}_t - \eta_t \frac{1}{M} \sum_{i=1}^{M} e_{t+1,i} && \text{Apply } \tilde{e}_{t+1} = \tilde{p}_t - \mathcal{C}(\tilde{p}_t) \\
&= x_t - \eta_t \frac{1}{M} \sum_{i=1}^{M} (\Delta_{t,i} + e_{t+1,i}) - \eta_{t-1} \tilde{e}_t && \text{Apply } \tilde{p}_t = \frac{1}{M} \sum_{i=1}^{M} \Delta_{t,i} + \frac{\eta_{t-1}}{\eta_t} \tilde{e}_t \\
&= x_t - \eta_t \frac{1}{M} \sum_{i=1}^{M} p_{t,i} - \eta_{t-1} \tilde{e}_t && \text{Apply } e_{t+1,i} = p_{t,i} - \Delta_{t,i} \\
&= x_t - \eta_t \frac{1}{M} \sum_{i=1}^{M} z_{t,i} - \eta_{t-1} \frac{1}{M} \sum_{i=1}^{M} e_{t,i} - \eta_{t-1} \tilde{e}_t && \text{Apply } p_{t,i} = z_{t,i} + \frac{\eta_{t-1}}{\eta_t} e_{t,i} \\
&= \tilde{x}_t - \eta_t \frac{1}{M} \sum_{i=1}^{M} z_{t,i}.
\end{aligned}$$

The Lemmas 1 and 3 hold by substituting $z_{t,i} = g_{t,i}$ and $z_{t,i} = \mu m_{t,i} + g_{t,i}$, respectively.  $\square$

# C  Proof of Theorem 1

*Proof.* By the smoothness of the function $F$, we have

$$\begin{aligned}
&\mathbb{E}_t[F(\tilde{x}_{t+1})] \\
&\leq F(\tilde{x}_t) + \langle \nabla F(\tilde{x}_t), \mathbb{E}_t[\tilde{x}_{t+1} - \tilde{x}_t] \rangle + \frac{L}{2} \mathbb{E}_t \left[ \|\tilde{x}_{t+1} - \tilde{x}_t\|_2^2 \right] \\
&= F(\tilde{x}_t) - \eta_t \left\langle \nabla F(\tilde{x}_t), \mathbb{E}_t \left[ \frac{1}{M} \sum_{i=1}^{M} g_{t,i} \right] \right\rangle + \frac{L\eta_t^2}{2} \mathbb{E}_t \left[ \left\| \frac{1}{M} \sum_{i=1}^{M} g_{t,i} \right\|_2^2 \right] \\
&= F(\tilde{x}_t) - \eta_t \langle \nabla F(\tilde{x}_t), \nabla F(x_t) \rangle + \frac{L\eta_t^2}{2} \|\nabla F(x_t)\|_2^2 + \frac{L\eta_t^2}{2} \mathbb{E}_t \left[ \left\| \frac{1}{M} \sum_{i=1}^{M} g_{t,i} - \nabla F(x_t) \right\|_2^2 \right] \\
&\leq F(\tilde{x}_t) - \eta_t \langle \nabla F(\tilde{x}_t), \nabla F(x_t) \rangle + \frac{L\eta_t^2}{2} \|\nabla F(x_t)\|_2^2 + \frac{L\eta_t^2 \sigma^2}{2M}
\end{aligned}$$

where in the second equality we use Lemma 1, and the second-to-last inequality follows the fact $\mathbb{E}[\|x - \mathbb{E}[x]\|_2^2] = \mathbb{E}[\|x\|_2^2] - \|\mathbb{E}[x]\|_2^2$. In the last inequality, we use the variance bound of the

mini-batch gradient, i.e., $\mathbb{E}_t\left[\left\|\frac{1}{M}\sum_{i=1}^M g_{t,i} - \nabla F(x_t)\right\|_2^2\right] \le \frac{\sigma^2}{M}$. Then, we get

$\mathbb{E}_t[F(\tilde{x}_{t+1})]$

$$
\begin{aligned}
&\le\; F(\tilde{x}_t) - \eta_t\langle\nabla F(x_t), \nabla F(x_t)\rangle + \frac{L\eta_t^2}{2}\|\nabla F(x_t)\|_2^2 + \frac{L\eta_t^2\sigma^2}{2M} + \eta_t\langle\nabla F(x_t) - \nabla F(\tilde{x}_t), \nabla F(x_t)\rangle\\
&=\; F(\tilde{x}_t) - \eta_t\left(1 - \frac{L\eta_t}{2}\right)\|\nabla F(x_t)\|_2^2 + \frac{L\eta_t^2\sigma^2}{2M} + \eta_t\langle\nabla F(x_t) - \nabla F(\tilde{x}_t), \nabla F(x_t)\rangle\\
&\le\; F(\tilde{x}_t) - \eta_t\left(1 - \frac{L\eta_t}{2}\right)\|\nabla F(x_t)\|_2^2 + \frac{L\eta_t^2\sigma^2}{2M} + \frac{\eta_t\rho}{2}\|\nabla F(x_t)\|_2^2 + \frac{\eta_t}{2\rho}\|\nabla F(x_t) - \nabla F(\tilde{x}_t)\|_2^2\\
&=\; F(\tilde{x}_t) - \eta_t\left(1 - \frac{L\eta_t + \rho}{2}\right)\|\nabla F(x_t)\|_2^2 + \frac{L\eta_t^2\sigma^2}{2M} + \frac{\eta_t}{2\rho}\|\nabla F(x_t) - \nabla F(\tilde{x}_t)\|_2^2\\
&\le\; F(\tilde{x}_t) - \eta_t\left(1 - \frac{L\eta_t + \rho}{2}\right)\|\nabla F(x_t)\|_2^2 + \frac{L\eta_t^2\sigma^2}{2M} + \frac{\eta_t L^2}{2\rho}\|x_t - \tilde{x}_t\|_2^2\\
&=\; F(\tilde{x}_t) - \eta_t\left(1 - \frac{L\eta_t + \rho}{2}\right)\|\nabla F(x_t)\|_2^2 + \frac{L\eta_t^2\sigma^2}{2M} + \frac{\eta_t\eta_{t-1}^2 L^2}{2\rho}\left\|\tilde{e}_t + \frac{1}{M}\sum_{i=1}^M e_{t,i}\right\|_2^2,
\end{aligned}
$$

where the second inequality follows from Young's inequality with $\rho > 0$. The last inequality follows from the smoothness of the function $F$. Let $\rho = 1/2$. Taking total expectation and using Lemma 6 with $\mu = 0$, we get

$\mathbb{E}_t[F(\tilde{x}_{t+1})]$

$$
\le\; \mathbb{E}[F(\tilde{x}_t)] - \eta_t\left(\frac{3}{4} - \frac{L\eta_t}{2}\right)\mathbb{E}[\|\nabla F(x_t)\|_2^2] + \frac{L\eta_t^2\sigma^2}{2M} + \frac{8L^2\eta_t\eta_{t-1}^2(1-\delta)G^2}{\delta^2}\left[1 + \frac{16}{\delta^2}\right].
$$

Assume that $\eta_t < 3/(2L)$ for all $t$. Rearranging the terms, taking summation, and dividing by $\sum_{k=0}^{T-1}\frac{\eta_k}{4}(3 - 2L\eta_k)$ gives

$$
\begin{aligned}
&\frac{1}{\sum_{k=0}^{T-1}\eta_k(3 - 2L\eta_k)}\sum_{t=0}^{T-1}\eta_t(3 - 2L\eta_t)\,\mathbb{E}\left[\|\nabla F(x_t)\|_2^2\right]\\
&\le\; \frac{4}{\sum_{k=0}^{T-1}\eta_k(3 - 2L\eta_k)}\sum_{t=0}^{T-1}\mathbb{E}[F(\tilde{x}_t) - F(\tilde{x}_{t+1})] + \frac{2L\sigma^2}{M}\sum_{t=0}^{T-1}\frac{\eta_t^2}{\sum_{k=0}^{T-1}\eta_k(3 - 2L\eta_k)}\\
&\quad + \frac{32L^2(1-\delta)G^2}{\delta^2}\left[1 + \frac{16}{\delta^2}\right]\sum_{t=0}^{T-1}\frac{\eta_t\eta_{t-1}^2}{\sum_{k=0}^{T-1}\eta_k(3 - 2L\eta_k)}\\
&\le\; \frac{4}{\sum_{k=0}^{T-1}\eta_k(3 - 2L\eta_k)}[F(x_0) - F_*] + \frac{2L\sigma^2}{M}\sum_{t=0}^{T-1}\frac{\eta_t^2}{\sum_{k=0}^{T-1}\eta_k(3 - 2L\eta_k)}\\
&\quad + \frac{32L^2(1-\delta)G^2}{\delta^2}\left[1 + \frac{16}{\delta^2}\right]\sum_{t=0}^{T-1}\frac{\eta_t\eta_{t-1}^2}{\sum_{k=0}^{T-1}\eta_k(3 - 2L\eta_k)}.
\end{aligned}
$$

Let $o \in \{0, \ldots, T-1\}$ be an index such that

$$
P(o = k) = \frac{\eta_k(3 - 2L\eta_k)}{\sum_{t=0}^{T-1}\eta_t(3 - 2L\eta_t)}.
$$

Then, we have

$$
\mathbb{E}[\|\nabla F(x_o)\|_2^2] = \frac{1}{\sum_{k=0}^{T-1}\eta_k(3 - 2L\eta_k)}\sum_{t=0}^{T-1}\eta_t(3 - 2L\eta_t)\,\mathbb{E}\left[\|\nabla F(x_t)\|_2^2\right],
$$

which concludes the results. $\qquad\square$

## D    Proof of Corollary 1

*Proof.* Let $\eta_t = \eta$ for all $t$, we have

$$
\mathbb{E}[\|\nabla F(x_o)\|_2^2] \leq \frac{4}{\eta\,(3-2L\eta)\,T}[F(x_0) - F_*] + \frac{2L\eta\sigma^2}{(3-2L\eta)\,M}
$$
$$
+ \frac{32L^2\eta^2(1-\delta)G^2}{(3-2L\eta)\,\delta^2}\left[1 + \frac{16}{\delta^2}\right]. \tag{3}
$$

Let $\eta = \min\left(\dfrac{1}{2L}, \dfrac{\gamma}{\frac{\sqrt{T}}{\sqrt{M}} + \frac{(1-\delta)^{1/3}}{\delta^{2/3}}\left(1+\frac{16}{\delta^2}\right)^{1/3}T^{1/3}}\right)$ for some $\gamma > 0$, then $3 - 2L\eta \geq 2$. Substituting this into (3), we get

$$
\mathbb{E}[\|\nabla F(x_o)\|_2^2]
$$
$$
\leq \frac{2}{\eta T}[F(x_0) - F_*] + \frac{L\eta\sigma^2}{M} + \frac{16L^2\eta^2(1-\delta)G^2}{\delta^2}\left[1 + \frac{16}{\delta^2}\right]
$$
$$
\leq \frac{2}{T}\max\left(2L, \frac{\sqrt{T}}{\gamma\sqrt{M}} + \frac{(1-\delta)^{1/3}}{\gamma\delta^{2/3}}\left[1+\frac{16}{\delta^2}\right]^{1/3}T^{1/3}\right)[F(x_0) - F_*]
$$
$$
+ \frac{L\eta\sigma^2}{M} + \frac{16L^2\eta^2(1-\delta)G^2}{\delta^2}\left[1 + \frac{16}{\delta^2}\right]
$$
$$
\leq \frac{4L}{T}[F(x_0) - F_*] + \left[\frac{2}{\gamma\sqrt{MT}} + \frac{2(1-\delta)^{1/3}}{\gamma\delta^{2/3}T^{2/3}}\left[1+\frac{16}{\delta^2}\right]^{1/3}\right][F(x_0) - F_*]
$$
$$
+ \frac{L\gamma\sigma^2}{\sqrt{MT}} + \frac{16L^2\gamma^2(1-\delta)^{1/3}G^2}{\delta^{2/3}T^{2/3}}\left[1 + \frac{16}{\delta^2}\right]^{1/3}
$$
$$
= \frac{4L}{T}[F(x_0) - F_*] + \left[\frac{2}{\gamma}[F(x_0) - F_*] + L\gamma\sigma^2\right]\frac{1}{\sqrt{MT}}
$$
$$
+ \frac{2(1-\delta)^{1/3}[\frac{1}{\gamma}[F(x_0) - F_*] + 8L^2\gamma^2 G^2]}{\delta^{2/3}T^{2/3}}\left[1 + \frac{16}{\delta^2}\right]^{1/3}.
$$

The bound on full-precision distributed SGD follows similar proof. For completeness, we present proof here. By the smoothness of the function $F$, we have

$$
\mathbb{E}_t[F(x_{t+1})]
$$
$$
\leq F(x_t) + \langle \nabla F(x_t), \mathbb{E}_t[x_{t+1} - x_t]\rangle + \frac{L}{2}\mathbb{E}_t\left[\|x_{t+1} - x_t\|_2^2\right]
$$
$$
= F(x_t) - \eta_t\left\langle \nabla F(x_t), \mathbb{E}_t\left[\frac{1}{M}\sum_{i=1}^{M} g_{t,i}\right]\right\rangle + \frac{L\eta_t^2}{2}\mathbb{E}_t\left[\left\|\frac{1}{M}\sum_{i=1}^{M} g_{t,i}\right\|_2^2\right]
$$
$$
= F(\tilde{x}_t) - \eta_t\left(1 - \frac{L\eta_t}{2}\right)\|\nabla F(x_t)\|_2^2 + \frac{L\eta_t^2}{2}\mathbb{E}_t\left[\left\|\frac{1}{M}\sum_{i=1}^{M} g_{t,i} - \nabla F(x_t)\right\|_2^2\right]
$$
$$
\leq F(x_t) - \eta_t\left(1 - \frac{L\eta_t}{2}\right)\|\nabla F(x_t)\|_2^2 + \frac{L\eta_t^2\sigma^2}{2M}.
$$

Let $\eta_t = \eta$. Taking total expectation, rearranging terms, and averaging over $T$, we obtain

$$
\mathbb{E}\left[\|\nabla F(x_o)\|_2^2\right] \leq \frac{2}{\eta\,(2-L\eta)\,T}[F(x_0) - F_*] + \frac{L\eta\sigma^2}{(2-L\eta)M}.
$$

Substituting $\eta = \min\left(\frac{1}{2L}, \frac{\gamma\sqrt{M}}{\sqrt{T}}\right)$, we get

$$
\begin{aligned}
\mathbb{E}\left[\|\nabla F(x_o)\|_2^2\right] &\leq \frac{4}{3\eta T}[F(x_0) - F_*] + \frac{2L\eta\sigma^2}{3M} \\
&\leq \frac{8L}{3T}[F(x_0) - F_*] + \frac{4}{3\gamma\sqrt{T}}[F(x_0) - F_*] + \frac{2L\gamma\sigma^2}{3M\sqrt{T}} \\
&= \frac{8L}{3T}[F(x_0) - F_*] + \left[\frac{2}{\gamma}[F(x_0) - F_*] + L\gamma\sigma^2\right]\frac{2}{3\sqrt{MT}}.
\end{aligned}
$$

$\square$

## E   Proof of Corollary 2

*Proof.* Let $\eta_t = \dfrac{\gamma}{\frac{((t+1)T)^{1/4}}{\sqrt{M}} + \frac{(1-\delta)^{1/3}}{\delta^{2/3}}\left(1 + \frac{16}{\delta^2}\right)^{1/3}T^{1/3}}$. The following implies that $\eta_t \leq 1/(2L)$ for all $0 \leq t \leq T - 1$.

$$
T \geq 16L^4\gamma^4M^2.
$$

Then, we have

$$
\begin{aligned}
\sum_{t=0}^{T-1} \eta_t &= \gamma\sum_{t=0}^{T-1} \frac{1}{\frac{((t+1)T)^{1/4}}{\sqrt{M}} + \frac{(1-\delta)^{1/3}}{\delta^{2/3}}\left(1 + \frac{16}{\delta^2}\right)^{1/3}T^{1/3}} \\
&\geq \gamma\sum_{t=0}^{T-1} \frac{1}{\frac{\sqrt{T}}{\sqrt{M}} + \frac{(1-\delta)^{1/3}}{\delta^{2/3}}\left(1 + \frac{16}{\delta^2}\right)^{1/3}T^{1/3}} \\
&= \frac{1}{\frac{1}{\gamma\sqrt{MT}} + \frac{(1-\delta)^{1/3}}{\gamma\delta^{2/3}T^{2/3}}\left(1 + \frac{16}{\delta^2}\right)^{1/3}}.
\end{aligned}
$$

Using the fact that $\sum_{t=1}^{T} t^{\alpha-1} \leq \int_0^T x^{\alpha-1}dx = \frac{T^\alpha}{\alpha}$, for any $0 < \alpha < 1$, we have

$$
\begin{aligned}
\sum_{t=0}^{T-1} \eta_t^2 &\leq \frac{\gamma^2 M}{\sqrt{T}}\sum_{t=1}^{T}\frac{1}{\sqrt{t}} \leq 2\gamma^2 M, \\
\sum_{t=0}^{T-1} \eta_t\eta_{t-1}^2 &= \sum_{t=1}^{T-1} \eta_t\eta_{t-1}^2 \leq \sum_{t=1}^{T-1}\eta_{t-1}^3 \leq \frac{\gamma^3}{\frac{(1-\delta)}{\delta^2}\left(1 + \frac{16}{\delta^2}\right)}.
\end{aligned}
$$

Substituting the above results into Theorem 1, we obtain

$$
\begin{aligned}
&\mathbb{E}\left[\|\nabla F(x_o)\|_2^2\right] \\
&\leq \left[\frac{1}{\gamma\sqrt{MT}} + \frac{(1-\delta)^{1/3}}{\gamma\delta^{2/3}T^{2/3}}\left(1 + \frac{16}{\delta^2}\right)^{1/3}\right]2[F(x_0) - F_*] \\
&\quad + \left[\frac{1}{\gamma\sqrt{MT}} + \frac{(1-\delta)^{1/3}}{\gamma\delta^{2/3}T^{2/3}}\left(1 + \frac{16}{\delta^2}\right)^{1/3}\right]2L\gamma^2\sigma^2 \\
&\quad + \left[\frac{1}{\gamma\sqrt{MT}} + \frac{(1-\delta)^{1/3}}{\gamma\delta^{2/3}T^{2/3}}\left(1 + \frac{16}{\delta^2}\right)^{1/3}\right]16L^2\gamma^3 G^2 \\
&= 2\left[\frac{1}{\sqrt{MT}} + \frac{(1-\delta)^{1/3}}{\delta^{2/3}T^{2/3}}\left(1 + \frac{16}{\delta^2}\right)^{1/3}\right]\left[\frac{1}{\gamma}[F(x_0) - F_*] + L\gamma\sigma^2 + 8L^2\gamma^2 G^2\right].
\end{aligned}
$$

Similarly, let $\eta_t = \frac{\gamma\sqrt{t+1}}{\frac{T}{\sqrt{M}} + \frac{(1-\delta)^{1/3}}{\delta^{2/3}}\left(1+\frac{16}{\delta^2}\right)^{1/3}T^{5/6}}$. We obtain

$$
\begin{aligned}
\sum_{t=0}^{T-1}\eta_t &= \gamma\sum_{t=0}^{T-1}\frac{\sqrt{t+1}}{\frac{T}{\sqrt{M}} + \frac{(1-\delta)^{1/3}}{\delta^{2/3}}\left(1+\frac{16}{\delta^2}\right)^{1/3}T^{5/6}}\\
&= \gamma\sum_{t=1}^{T}\frac{\sqrt{t}}{\frac{T}{\sqrt{M}} + \frac{(1-\delta)^{1/3}}{\delta^{2/3}}\left(1+\frac{16}{\delta^2}\right)^{1/3}T^{5/6}}\\
&\geq \gamma\int_{0}^{T}\frac{\sqrt{x}}{\frac{T}{\sqrt{M}} + \frac{(1-\delta)^{1/3}}{\delta^{2/3}}\left(1+\frac{16}{\delta^2}\right)^{1/3}T^{5/6}}dx\\
&= \frac{2T^{3/2}}{\frac{3T}{\gamma\sqrt{M}} + \frac{3(1-\delta)^{1/3}}{\gamma\delta^{2/3}}\left(1+\frac{16}{\delta^2}\right)^{1/3}T^{5/6}}.
\end{aligned}
$$

Using the fact that $\sum_{t=1}^{T}t^\alpha \leq \int_{1}^{T+1}x^\alpha dx \leq \frac{(T+1)^{\alpha+1}}{\alpha+1}$ for any $\alpha > 0$, we also have

$$
\sum_{t=0}^{T-1}\eta_t^2 \leq \frac{\gamma^2 M}{T^2}\sum_{t=1}^{T}t = \frac{\gamma^2 M(T+1)}{2T},
$$

$$
\sum_{t=0}^{T-1}\eta_t\eta_{t-1}^2 = \sum_{t=0}^{T-1}\eta_t\eta_{t-1}^2 \leq \sum_{t=1}^{T-1}\eta_t^3 \leq \frac{2\gamma^3(T+1)^{5/2}}{5\frac{(1-\delta)}{\delta^2}\left(1+\frac{16}{\delta^2}\right)T^{5/2}}.
$$

Assuming that $T \geq 4L^2\gamma^2 M$, we have $\eta_t \leq 1/(2L)$ for all $0 \leq t \leq T-1$. Substituting the above results into Theorem 1, we obtain

$$
\mathbb{E}\left[\|\nabla F(x_o)\|_2^2\right]
$$
$$
\begin{aligned}
&\leq \left[\frac{1}{\gamma\sqrt{MT}} + \frac{(1-\delta)^{1/3}}{\gamma\delta^{2/3}T^{2/3}}\left(1+\frac{16}{\delta^2}\right)^{1/3}\right]3[F(x_0)-F_*]\\
&\quad + \left[\frac{1}{\gamma\sqrt{MT}} + \frac{(1-\delta)^{1/3}}{\gamma\delta^{2/3}T^{2/3}}\left(1+\frac{16}{\delta^2}\right)^{1/3}\right]\frac{3L\gamma^2\sigma^2(T+1)}{4T}\\
&\quad + \left[\frac{1}{\gamma\sqrt{MT}} + \frac{(1-\delta)^{1/3}}{\gamma\delta^{2/3}T^{2/3}}\left(1+\frac{16}{\delta^2}\right)^{1/3}\right]\frac{48L^2\gamma^3 G^2(T+1)^{5/2}}{5T^{5/2}}\\
&= 3\left[\frac{1}{\sqrt{MT}} + \frac{(1-\delta)^{1/3}}{\delta^{2/3}T^{2/3}}\left(1+\frac{16}{\delta^2}\right)^{1/3}\right]\left[\frac{1}{\gamma}[F(x_0)-F_*] + \frac{L\gamma\sigma^2(T+1)}{4T} + \frac{16L^2\gamma^2 G^2(T+1)^{5/2}}{5T^{5/2}}\right].
\end{aligned}
$$

$\square$

# F  Proof of Proposition 1

*Proof.*

$$
\begin{aligned}
\|\mathcal{C}_B(v) - v\|_2^2 &= \sum_{b=1}^{B}\left\|\frac{\|v_{\mathcal{G}_b}\|_1}{d_b}\mathrm{sign}(v_{\mathcal{G}_b}) - v_{\mathcal{G}_b}\right\|_2^2\\
&= \sum_{b=1}^{B}\left[\frac{\|v_{\mathcal{G}_b}\|_1^2}{d_b} - 2\frac{\|v_{\mathcal{G}_b}\|_1^2}{d_b} + \|v_{\mathcal{G}_b}\|_2^2\right]\\
&= \sum_{b=1}^{B}\left(1 - \frac{\|v_{\mathcal{G}_b}\|_1^2}{d_b\|v_{\mathcal{G}_b}\|_2^2}\right)\|v_{\mathcal{G}_b}\|_2^2\\
&\leq \left(1 - \min_{b\in[B]}\frac{\|v_{\mathcal{G}_b}\|_1^2}{d_b\|v_{\mathcal{G}_b}\|_2^2}\right)\|v\|_2^2.
\end{aligned}
$$

$\square$

## G  Proof of Theorem 2

We first introduce the following Lemmas.

**Lemma 5.** *For any* $i \in [M]$, *we have*

$$\mathbb{E}\left[\|\mu m_{t,i} + g_{t,i}\|_2^2\right] \leq \frac{G^2}{1-\mu}.$$

*Proof.*

$$
\begin{aligned}
\mathbb{E}\left[\|\mu m_{t,i} + g_{t,i}\|_2^2\right] &= \mathbb{E}\left[\left\|\sum_{k=1}^{t} \mu^{t-k+1} g_{k,i} + g_{t,i}\right\|_2^2\right] \\
&= \left(\sum_{k=1}^{t} \mu^{t-k+1} + 1\right)^2 \mathbb{E}\left[\left\|\frac{\sum_{k=1}^{t} \mu^{t-k+1} g_{k,i} + g_{t,i}}{\sum_{k=1}^{t} \mu^{t-k+1} + 1}\right\|_2^2\right] \\
&\leq \left(\sum_{k=1}^{t} \mu^{t-k+1} + 1\right)\left(\sum_{k=1}^{t} \mu^{t-k+1}\mathbb{E}\left[\|g_{k,i}\|_2^2\right] + \mathbb{E}\left[\|g_{t,i}\|_2^2\right]\right) \\
&\leq \left(\sum_{k=1}^{t} \mu^{t-k+1} + 1\right)^2 G^2 \\
&\leq \frac{G^2}{(1-\mu)^2},
\end{aligned}
$$

where in the first inequality we use Jensen's inequality. In the second-to-last equality, we apply Assumptions 2 and 3. The last inequality follows from the sum of a geometric series. $\square$

**Lemma 6.** *For any* $t \geq 0$, *we have*

$$\mathbb{E}\left[\left\|\tilde{e}_t + \frac{1}{M}\sum_{i=1}^{M} e_{t,i}\right\|_2^2\right] \leq \frac{8(1-\delta)G^2}{\delta^2(1-\mu)^2}\left[1 + \frac{16}{\delta^2}\right].$$

*Proof.* When $t = 0$, the bound trivially holds as $\tilde{e}_0 = 0$ and $e_{0,i} = 0$ for all $i$. Using $(a+b)^2 \leq 2a^2 + 2b^2$, we get

$$
\begin{aligned}
\left\|\tilde{e}_{t+1} + \frac{1}{M}\sum_{i=1}^{M} e_{t+1,i}\right\|_2^2 &\leq 2\|\tilde{e}_{t+1}\|_2^2 + 2\left\|\frac{1}{M}\sum_{i=1}^{M} e_{t+1,i}\right\|_2^2 \\
&\leq 2\|\tilde{e}_{t+1}\|_2^2 + \frac{2}{m}\sum_{i=1}^{M}\|e_{t+1,i}\|_2^2, \quad \forall t \geq 0. \tag{4}
\end{aligned}
$$

Now, we can consider two terms separately. For the second term, we have

$$
\frac{1}{M}\sum_{i=1}^{M}\mathbb{E}\left[\|e_{t+1,i}\|_2^2\right] = \frac{1}{M}\sum_{i=1}^{M}\mathbb{E}\left[\|\mathcal{C}(p_{t,i}) - p_{t,i}\|_2^2\right]
$$

$$
\leq (1-\delta)\frac{1}{M}\sum_{i=1}^{M}\mathbb{E}\left[\|p_{t,i}\|_2^2\right] \tag{5}
$$

$$
= (1-\delta)\frac{1}{M}\sum_{i=1}^{M}\mathbb{E}\left[\|e_{t,i} + \mu m_{t,i} + g_{t,i}\|_2^2\right]
$$

$$
\leq (1-\delta)(1+\beta)\frac{1}{M}\sum_{i=1}^{M}\mathbb{E}\left[\|e_{t,i}\|_2^2\right] + (1-\delta)(1+1/\beta)\frac{1}{M}\sum_{i=1}^{M}\mathbb{E}\left[\|\mu m_{t,i} + g_{t,i}\|_2^2\right]
$$

$$
\leq (1-\delta)(1+\beta)\frac{1}{M}\sum_{i=1}^{M}\mathbb{E}\left[\|e_{t,i}\|_2^2\right] + (1-\delta)(1+1/\beta)\frac{G^2}{(1-\mu)^2}
$$

$$
\leq \sum_{k=0}^{t}[(1-\delta)(1+\beta)]^{t-k}(1-\delta)(1+1/\beta)\frac{G^2}{(1-\mu)^2}
$$

$$
\leq \frac{(1-\delta)(1+1/\beta)}{1-(1-\delta)(1+\beta)}\frac{G^2}{(1-\mu)^2} = \frac{(1-\delta)(1+1/\beta)}{\delta - \beta(1-\delta)}\frac{G^2}{(1-\mu)^2},
$$

where the first inequality follows from the definition of the compressor $\mathcal{C}$. The second inequality follows from Young's inequality with any $\beta > 0$, and the third inequality follows from Lemma 5. The third equality follows from the definition of $p_{t,i}$ and the assumption $\eta_t = \eta$. The last inequality follows from the sum of a geometric series. Let $\beta = \frac{\delta}{2(1-\delta)}$, then $1 + 1/\beta = (2-\delta)/\delta \leq 2/\delta$. We get

$$
\frac{1}{M}\sum_{i=1}^{M}\mathbb{E}\left[\|e_{t+1,i}\|_2^2\right] \leq \frac{(1-\delta)(1+1/\beta)}{\delta - \beta(1-\delta)}\frac{G^2}{(1-\mu)^2} = \frac{2(1-\delta)(1+1/\beta)}{\delta(1-\mu)^2}G^2 \leq \frac{4(1-\delta)}{\delta^2(1-\mu)^2}G^2. \tag{6}
$$

Then, the first term can be bounded as

$$
\mathbb{E}\left[\|\tilde{e}_{t+1}\|_2^2\right] = \mathbb{E}\left[\|\mathcal{C}(\tilde{p}_t) - \tilde{p}_t\|_2^2\right] \leq (1-\delta)\mathbb{E}\left[\|\tilde{p}_t\|_2^2\right]
$$

$$
= (1-\delta)\mathbb{E}\left[\left\|\frac{1}{M}\sum_{i=1}^{M}\Delta_{t,i} + \tilde{e}_t\right\|_2^2\right]
$$

$$
\leq (1-\delta)(1+\beta)\mathbb{E}\left[\|\tilde{e}_t\|_2^2\right] + (1-\delta)(1+1/\beta)\mathbb{E}\left[\left\|\frac{1}{M}\sum_{i=1}^{M}\Delta_{t,i}\right\|_2^2\right]
$$

$$
\leq (1-\delta)(1+\beta)\mathbb{E}\left[\|\tilde{e}_t\|_2^2\right] + 2(1-\delta)(1+1/\beta)\mathbb{E}\left[\left\|\frac{1}{M}\sum_{i=1}^{M}\Delta_{t,i} - \frac{1}{M}\sum_{i=1}^{M}p_{t,i}\right\|_2^2\right]
$$

$$
+ 2(1-\delta)(1+1/\beta)\mathbb{E}\left[\left\|\frac{1}{M}\sum_{i=1}^{M}p_{t,i}\right\|_2^2\right]
$$

$$
\leq (1-\delta)(1+\beta)\mathbb{E}\left[\|\tilde{e}_t\|_2^2\right] + 2(1-\delta)^2(1+1/\beta)\frac{1}{M}\sum_{i=1}^{M}\mathbb{E}\left[\|p_{t,i}\|_2^2\right]
$$

$$
+ 2(1-\delta)(1+1/\beta)\frac{1}{M}\sum_{i=1}^{M}\mathbb{E}\left[\|p_{t,i}\|_2^2\right]
$$

$$
= (1-\delta)(1+\beta)\mathbb{E}\left[\|\tilde{e}_t\|_2^2\right] + 2(1-\delta)(2-\delta)(1+1/\beta)\frac{1}{M}\sum_{i=1}^{M}\mathbb{E}\left[\|p_{t,i}\|_2^2\right]. \tag{7}
$$

Combining (5), (6), we have $\frac{1}{M} \sum_{i=1}^{M} \mathbb{E}\left[\|p_{t,i}\|_2^2\right] \leq \frac{4}{\delta^2(1-\mu)^2}G^2$. Substituting it into (7), we get

$$
\begin{aligned}
\mathbb{E}&\left[\|\tilde{e}_{t+1}\|_2^2\right] \\
&\leq \quad (1-\delta)(1+\beta)\mathbb{E}\left[\|\tilde{e}_t\|_2^2\right] + \frac{8(1-\delta)(2-\delta)(1+1/\beta)}{\delta^2(1-\mu)^2}G^2 \\
&\leq \quad \sum_{k=0}^{t}[(1-\delta)(1+\beta)]^{t-k}\frac{8(1-\delta)(2-\delta)(1+1/\beta)}{\delta^2(1-\mu)^2}G^2 \\
&\leq \quad \frac{8(1-\delta)(2-\delta)(1+1/\beta)}{\delta^2(1-(1-\delta)(1+\beta))(1-\mu)^2}G^2 \\
&= \quad \frac{8(1-\delta)(2-\delta)(1+1/\beta)}{\delta^2(\delta-\beta(1-\delta))(1-\mu)^2}G^2 \\
&= \quad \frac{16(1-\delta)(2-\delta)(1+1/\beta)}{\delta^3(1-\mu)^2}G^2 \\
&= \quad \frac{32(1-\delta)(2-\delta)}{\delta^4(1-\mu)^2}G^2 \\
&\leq \quad \frac{64(1-\delta)}{\delta^4(1-\mu)^2}G^2. \quad\quad (8)
\end{aligned}
$$

Then, combining (4), (6) and (8), we obtain

$$
\mathbb{E}\left[\left\|\tilde{e}_{t+1} + \frac{1}{M}\sum_{i=1}^{M}e_{t+1,i}\right\|_2^2\right] \leq \frac{8(1-\delta)G^2}{\delta^2(1-\mu)^2}\left[1+\frac{16}{\delta^2}\right].
$$

$\square$

*Proof.* In the sequel, we assume $\eta_t = \eta$ for some $\eta > 0$. Let us introduce the following virtual iterate:

$$
z_t \quad = \quad \tilde{x}_t - \frac{\eta\mu^2}{1-\mu}\frac{1}{M}\sum_{i=1}^{M}m_{t-1,i},
$$

where $\tilde{x}_t$ is defined in Lemma 3 . Then, it satisfies the following recurrence:

$$
\begin{aligned}
z_{t+1} \quad &= \quad \tilde{x}_{t+1} - \frac{\eta\mu^2}{1-\mu}\frac{1}{M}\sum_{i=1}^{M}m_{t,i} \\
&= \quad \tilde{x}_t - \eta\frac{1}{M}\sum_{i=1}^{M}(\mu m_{t,i}+g_{t,i}) - \frac{\eta\mu^2}{1-\mu}\frac{1}{M}\sum_{i=1}^{M}m_{t,i} \\
&= \quad \tilde{x}_t - \frac{\eta\mu}{1-\mu}\frac{1}{M}\sum_{i=1}^{M}m_{t,i} - \eta\frac{1}{M}\sum_{i=1}^{M}g_{t,i} \\
&= \quad \tilde{x}_t - \frac{\eta\mu^2}{1-\mu}\frac{1}{M}\sum_{i=1}^{M}m_{t-1,i} - \frac{\eta\mu}{1-\mu}\frac{1}{M}\sum_{i=1}^{M}g_{t,i} - \eta\frac{1}{M}\sum_{i=1}^{M}g_{t,i} \\
&= \quad z_t - \frac{\eta}{1-\mu}\frac{1}{M}\sum_{i=1}^{M}g_{t,i}.
\end{aligned}
$$

By the smoothness of the function $F$, we get

$\mathbb{E}_t[F(z_{t+1})]$

$$\leq F(z_t) + \langle \nabla F(z_t), \mathbb{E}_t[z_{t+1} - z_t] \rangle + \frac{L}{2} \mathbb{E}_t[\|z_{t+1} - z_t\|_2^2]$$

$$= F(z_t) - \frac{\eta}{1-\mu} \left\langle \nabla F(z_t), \mathbb{E}_t \left[ \frac{1}{M} \sum_{i=1}^{M} g_{t,i} \right] \right\rangle + \frac{L\eta^2}{2(1-\mu)^2} \mathbb{E}_t \left[ \left\| \frac{1}{M} \sum_{i=1}^{M} g_{t,i} \right\|_2^2 \right]$$

$$= F(z_t) - \frac{\eta}{1-\mu} \langle \nabla F(z_t), \nabla F(x_t) \rangle + \frac{L\eta^2}{2(1-\mu)^2} \left[ \|\nabla F(x_t)\|_2^2 + \mathbb{E}_t \left[ \left\| \frac{1}{M} \sum_{i=1}^{M} g_{t,i} - \nabla F(x_t) \right\|_2^2 \right] \right]$$

$$\leq F(z_t) - \frac{\eta}{1-\mu} \langle \nabla F(z_t), \nabla F(x_t) \rangle + \frac{L\eta^2}{2(1-\mu)^2} \|\nabla F(x_t)\|_2^2 + \frac{L\eta^2 \sigma^2}{2(1-\mu)^2 M}, \tag{9}$$

where the second-to-last equality follows from $\mathbb{E}[\|x - \mathbb{E}[x]\|_2^2] = \mathbb{E}[\|x\|_2^2] - \|\mathbb{E}[x]\|_2^2$. Then, we bound the second term $-\langle \nabla F(z_t), \nabla F(x_t) \rangle$.

$$-\langle \nabla F(z_t), \nabla F(x_t) \rangle = -\|\nabla F(x_t)\|_2^2 + \langle \nabla F(x_t) - \nabla F(z_t), \nabla F(x_t) \rangle$$

$$\leq -\left(1 - \frac{\rho}{2}\right) \|\nabla F(x_t)\|_2^2 + \frac{1}{2\rho} \|\nabla F(x_t) - \nabla F(z_t)\|_2^2 \tag{10}$$

for any $0 < \rho < 2$. Then, we have

$$\|\nabla F(x_t) - \nabla F(z_t)\|_2^2 \leq L^2 \|x_t - z_t\|_2^2$$

$$\leq 2L^2 \|x_t - \tilde{x}_t\|_2^2 + 2L^2 \|\tilde{x}_t - z_t\|_2^2$$

$$= 2L^2\eta^2 \left\| \tilde{e}_t + \frac{1}{M} \sum_{i=1}^{M} e_{t,i} \right\|_2^2 + \frac{2L^2\eta^2\mu^4}{(1-\mu)^2} \left\| \frac{1}{M} \sum_{i=1}^{M} m_{t-1,i} \right\|_2^2$$

$$\leq \frac{16L^2\eta^2(1-\delta)G^2}{\delta^2(1-\mu)^2} \left[ 1 + \frac{16}{\delta^2} \right] + \frac{2L^2\eta^2\mu^4}{(1-\mu)^2} \left\| \frac{1}{M} \sum_{i=1}^{M} m_{t-1,i} \right\|_2^2 \tag{11}$$

where in the last inequality we use Lemma 6. Let $A_{t-1} = \sum_{k=0}^{t-1} \mu^{t-1-k} = \frac{1-\mu^t}{1-\mu}$. Then, we bound the last term:

$$\left\| \frac{1}{M} \sum_{i=1}^{M} m_{t-1,i} \right\|_2^2 = A_{t-1}^2 \left\| \sum_{k=0}^{t-1} \frac{\mu^{t-1-k}}{A_{t-1}} \frac{1}{M} \sum_{i=1}^{M} g_{k,i} \right\|_2^2$$

$$\leq A_{t-1}^2 \sum_{k=0}^{t-1} \frac{\mu^{t-1-k}}{A_{t-1}} \left\| \frac{1}{M} \sum_{i=1}^{M} g_{k,i} \right\|_2^2$$

$$= A_{t-1} \sum_{k=0}^{t-1} \mu^{t-1-k} \left\| \frac{1}{M} \sum_{i=1}^{M} g_{k,i} \right\|_2^2$$

$$\leq \frac{1}{1-\mu} \sum_{k=0}^{t-1} \mu^{t-1-k} \left\| \frac{1}{M} \sum_{i=1}^{M} g_{k,i} \right\|_2^2, \tag{12}$$

where the first inequality follows from Jensen's inequality. Then, combining (9), (10), (11), and (12), we obtain

$\mathbb{E}_t[F(z_{t+1})]$

$$\leq F(z_t) - \left( \frac{\eta(2-\rho)}{2(1-\mu)} - \frac{L\eta^2}{2(1-\mu)^2} \right) \|\nabla F(x_t)\|_2^2 + \frac{L^2\eta^3\mu^4}{\rho(1-\mu)^4} \sum_{k=0}^{t-1} \mu^{t-1-k} \left\| \frac{1}{M} \sum_{i=1}^{M} g_{k,i} \right\|_2^2$$

$$+ \frac{L\eta^2\sigma^2}{2(1-\mu)^2 M} + \frac{8L^2\eta^3(1-\delta)G^2}{\rho\delta^2(1-\mu)^3} \left[ 1 + \frac{16}{\delta^2} \right].$$

Taking total expectation and telescoping this inequality from $0$ to $T-1$, we obtain

$$\left(\frac{\eta(2-\rho)}{2(1-\mu)} - \frac{L\eta^2}{2(1-\mu)^2}\right)\sum_{t=0}^{T-1}\mathbb{E}[\|\nabla F(x_t)\|_2^2]$$

$$\leq \quad \mathbb{E}[F(z_0)] - \mathbb{E}[F(z_T)] + \frac{L^2\eta^3\mu^4}{\rho(1-\mu)^4}\sum_{t=0}^{T-1}\sum_{k=0}^{t-1}\mu^{t-1-k}\mathbb{E}\left[\left\|\frac{1}{M}\sum_{i=1}^{M}g_{k,i}\right\|_2^2\right]$$

$$+\frac{L\eta^2\sigma^2 T}{2(1-\mu)^2 M} + \frac{8L^2\eta^3(1-\delta)G^2 T}{\rho\delta^2(1-\mu)^3}\left[1+\frac{16}{\delta^2}\right]$$

$$= \quad F(x_0) - \mathbb{E}[F(z_T)] + \frac{L^2\eta^3\mu^4}{\rho(1-\mu)^4}\sum_{t=0}^{T-1}\sum_{k=0}^{t-1}\mu^{t-1-k}\mathbb{E}\left[\|\nabla F(x_k)\|_2^2\right]$$

$$+\frac{L^2\eta^3\mu^4}{\rho(1-\mu)^4}\sum_{t=0}^{T-1}\sum_{k=0}^{t-1}\mu^{t-1-k}\mathbb{E}\left[\left\|\frac{1}{M}\sum_{i=1}^{M}g_{k,i} - \nabla F(x_k)\right\|_2^2\right]$$

$$+\frac{L\eta^2\sigma^2 T}{2(1-\mu)^2 M} + \frac{8L^2\eta^3(1-\delta)G^2 T}{\rho\delta^2(1-\mu)^3}\left[1+\frac{16}{\delta^2}\right]$$

$$\leq \quad F(x_0) - F_* + \frac{L^2\eta^3\mu^4}{\rho(1-\mu)^4}\sum_{t=0}^{T-1}\sum_{k=0}^{t-1}\mu^{t-1-k}\mathbb{E}\left[\|\nabla F(x_k)\|_2^2\right]$$

$$+\frac{L^2\eta^3\mu^4\sigma^2 T}{\rho(1-\mu)^5 M} + \frac{L\eta^2\sigma^2 T}{2(1-\mu)^2 M} + \frac{8L^2\eta^3(1-\delta)G^2 T}{\rho\delta^2(1-\mu)^3}\left[1+\frac{16}{\delta^2}\right].$$

Using double-sum trick, we get

$$\sum_{t=0}^{T-1}\sum_{k=0}^{t-1}\mu^{t-1-k}\mathbb{E}\left[\|\nabla F(x_k)\|_2^2\right] \quad = \quad \sum_{k=0}^{T-2}\sum_{t=k+1}^{T-1}\mu^{t-1-k}\mathbb{E}\left[\|\nabla F(x_k)\|_2^2\right]$$

$$\leq \quad \frac{1}{1-\mu}\sum_{k=0}^{T-2}\mathbb{E}\left[\|\nabla F(x_k)\|_2^2\right]$$

$$\leq \quad \frac{1}{1-\mu}\sum_{k=0}^{T-1}\mathbb{E}\left[\|\nabla F(x_k)\|_2^2\right].$$

Rearranging the terms, we get

$$\sum_{t=0}^{T-1}\left(\frac{\eta(2-\rho)}{2(1-\mu)} - \frac{L\eta^2}{2(1-\mu)^2} - \frac{L^2\eta^3\mu^4}{\rho(1-\mu)^5}\right)\mathbb{E}[\|\nabla F(x_t)\|_2^2]$$

$$\leq \quad F(x_0) - F_* + \frac{L^2\eta^3\mu^4\sigma^2 T}{\rho(1-\mu)^5 M} + \frac{L\eta^2\sigma^2 T}{2(1-\mu)^2 M} + \frac{8L^2\eta^3(1-\delta)G^2 T}{\rho\delta^2(1-\mu)^3}\left[1+\frac{16}{\delta^2}\right]. \quad (13)$$

Let $\eta \leq \frac{(2-\rho)(1-\mu)^2}{2L}$ and $\rho$ is selected such that $\rho \geq (2-\rho)\mu^3$, we get

$$\frac{\eta(2-\rho)}{2(1-\mu)} - \frac{L\eta^2}{2(1-\mu)^2} - \frac{L^2\eta^3\mu^4}{\rho(1-\mu)^5} \geq \frac{\eta(2-\rho)}{4(1-\mu)}. \quad (14)$$

Hence, combining (13) and (14), and dividing by $T$,

$$\frac{1}{T}\sum_{t=0}^{T-1}\mathbb{E}[\|\nabla F(x_t)\|_2^2] \leq \quad \frac{4(1-\mu)}{\eta(2-\rho)T}[F(x_0) - F_*] + \frac{2L\eta\sigma^2}{(2-\rho)(1-\mu)M}\left[1+\frac{2L\eta\mu^4}{\rho(1-\mu)^3}\right]$$

$$+\frac{32L^2\eta^2(1-\delta)G^2}{\rho(2-\rho)\delta^2(1-\mu)^2}\left[1+\frac{16}{\delta^2}\right].$$

Let $\rho = 1$ and $\mathbb{E}\left[\|\nabla F(x_o)\|_2^2\right] = \frac{1}{T}\sum_{t=0}^{T-1}\mathbb{E}[\|\nabla F(x_t)\|_2^2]$, we obtain the result. $\qquad\square$

# H Proof of Corollary 3

*Proof.* Let $\eta = \dfrac{\gamma}{\frac{\sqrt{T}}{\sqrt{M}} + \frac{(1-\delta)^{1/3}}{\delta^{2/3}}\left(1+\frac{16}{\delta^2}\right)^{1/3}T^{1/3}}$ for some $\gamma > 0$. As $T \geq \frac{4\gamma^2 L^2 M}{(1-\mu)^4}$, we have $\eta \leq \frac{(1-\mu)^2}{2L}$

and

$$
\mathbb{E}\left[\|\nabla F(x_o)\|_2^2\right]
$$

$$
\leq \quad \left[\frac{1}{\gamma\sqrt{MT}} + \frac{(1-\delta)^{1/3}}{\gamma\delta^{2/3}T^{2/3}}\left(1+\frac{16}{\delta^2}\right)^{1/3}\right]4(1-\mu)[F(x_0) - F_*]
$$

$$
+ \frac{2L\gamma\sigma^2}{(1-\mu)\sqrt{MT}}\left[1 + \frac{2L\gamma\mu^4\sqrt{M}}{(1-\mu)^3\sqrt{T}}\right] + \frac{32L^2\gamma^2(1-\delta)^{1/3}G^2}{\delta^{2/3}(1-\mu)^2 T^{2/3}}\left[1+\frac{16}{\delta^2}\right]^{1/3}
$$

$$
= \quad \left[\frac{2(1-\mu)}{\gamma}[F(x_0) - F_*] + \frac{L\gamma\sigma^2}{1-\mu}\right]\frac{2}{\sqrt{MT}} + \frac{4L^2\gamma^2\mu^4\sigma^2}{(1-\mu)^4 T}
$$

$$
+ \frac{4(1-\delta)^{1/3}\left[\frac{(1-\mu)}{\gamma}[F(x_0) - F_*] + \frac{8L^2\gamma^2 G^2}{(1-\mu)^2}\right]}{\delta^{2/3}T^{2/3}}\left[1+\frac{16}{\delta^2}\right]^{1/3}.
$$

$\square$