[Reviews · NeurIPS 2019]

Reviewer 1



The paper is clearly written, and the method seems to be working very nicely. The authors compare their method only to signSGD and signum, which as it seems lose a lot of accuracy compared to plain SGD and SGDM. However, looking at the QSGD paper, it looks like there isn't such a loss in accuracy against pure SGD. Can the authors compare their method to QSGD? The error feedback mechanism also appeared in [14] for a distributed setting. There is also the recent work below that applies very similar error-correction mechanism for QSGD: J.Wu, W. Huang, J. Huang, and T. Zhang. Error compensated quantized sgd and its applications to large-scale distributed optimization. In ICML, 2018. These works seem to apply the same error correction mechanisms. Can the authors distinguish their work from this one? Compare to it? Post rebuttal: the authors have addressed most of my concerns. In light of the rebuttal and the other reviews, I upgrade my score to 7.

Reviewer 2



Originality: The application of the compression and momentum to the parameter server is fairly original. The theorems established could be further adapted to other communication settings, including non-server ones such as binary tree allreduce or ring allreduce. Quality: Ideas and theorems are thoroughly well-explained. Experiments are done at a reasonable scale, though the number of repeat trials is too small to establish reproducibility, and it would have been nice to see more deep networks and datasets. Overall above average quality, but experiments could be more thorough. Clarity: The paper was well-written, very clear, and easy to follow. Significance: As mentioned in my earlier comment about originality, I think this paper lays stepping stones towards analyzing how compression and momentum can benefit communication settings beyond parameter server. High significance.

Reviewer 3



The novelty of this paper is limited and some important experiments are missing to support your claim. 1. Some citations of this paper are not proper while some important citations are missing. For example, when you review DNNs' success in different field, why [22] is cited? And you should also cite the following relevant papers: [1] Povey D, Zhang X, Khudanpur S. PARALLEL TRAINING OF DNNS WITH NATURAL GRA-DIENT AND PARAMETER AVERAGING[J]. arXiv preprint arXiv:1410.7455, 2014. [2] Chen K, Huo Q. Scalable training of deep learning machines by incremental block training with intra-block parallel optimization and blockwise model-update filtering[C]//2016 ieee international conference on acoustics, speech and signal processing (icassp). IEEE, 2016: 5880-5884. [3] Chen J, Pan X, Monga R, et al. Revisiting distributed synchronous SGD[J]. arXiv preprint arXiv:1604.00981, 2016. [4] Chen C Y, Choi J, Brand D, et al. Adacomp: Adaptive residual gradient compression for data-parallel distributed training[C]//Thirty-Second AAAI Conference on Artificial Intelligence. 2018. 2. Is it necessary to compress gradients passed from sever to worker? Since dist-EF-SGD is a synchronous algorithm, all_reduce can be used instead of reduce&broadcast. Leveraging all_reduce operation, every worker can play the role of server, and there is no need to send gradients from server to worker anymore. Have you compared the performance difference of these 2 implementations? 3. Your blockwise compression is based on the claim that "...gradients in a deep network typically have similar magnitudes in each layer". Can you provide experimental evidence of this claim? 4. Dist-EF-BlockSGD should be compared with Dist-EF-SGD to show the effect of blockwise compressor. 5. The introduction of nesterov momentum is one of your claims, but the experimental results show that dist-EF-SGD can benefit little from momentum, can you give an explanation?

[Author Response · NeurIPS 2019]

**Response to Reviewer 1**

**"compare their method to QSGD"**: Our ImageNet experiments are based on the code base of [4]. [4] already showed that (1) vanilla QSGD is worse than signum with majority vote; (2) when two-way compression is used, QSGD is still significantly worse. In this paper, we show that the proposed method outperforms signum, and thus outperforms QSGD.

**"also appeared in [14]"**: Ours is based on more advanced error feedback mechanism [9,16], which is more general and any $\delta$-approximate compressor can be used, while [14] is restricted to quantization methods. Moreover, [14] (with no theoretical guarantee) does not accumulate quantization errors as in MEM-SGD, EF-SGD and the proposed method.

**"distinguish their work from this one"**: Our dist-EF-SGD is better than Wu et al. (2018) as: (1) [Wu]: uses multi-bit quantization of QSGD; Ours: allows arbitrary compressors, including the unbiased quantization of QSGD (see lines 130-132). Thus, Algorithm 2 reduces to dist-EF-QSGD with two-way compression when QSGD's unbiased quantization is used; (2) [Wu]: past quantization errors are decayed exponentially. Thus, error feedback is limited to a small number of iterations; Ours: past quantization errors are decayed by a time-varying factor depending on stepsize. It can be shown from Corollary 2 that our stepsize choice ensures this factor to converge to one (so error neither explodes nor decays rapidly); (3) [Wu]: theoretical analysis only on quadratic objectives, with convergence to a neighborhood of optimal solution; Ours: global convergence to a stationary point for general nonconvex objectives. (4) [Wu]: uses two more hyper-parameters than ours; (5) Ours: employs Nesterov's momentum for better performance; (6) [Wu]: uses all-to-all broadcast (which may involve large network traffic and idle time); Ours: uses parameter-server architecture.

**Response to Reviewer 2**

"**fairly original**": Ours is the first that studies gradient compression with Nesterov's momentum in parameter-server, and shows theoretical guarantees. [11,14,20] only heuristically consider momentum, and [4] uses exponential moving average momentum without convergence analysis. [1,2,9,16,17] only study gradient compression with vanilla SGD.

"**averaged over only 3 repetitions**": Standard deviation is already very small. We will add more in the final version.

"**binary tree allreduce and ring allreduce**": Compression at server can be implemented between reduce and broadcast steps in tree allreduce, or between reduce-scatter and allgather steps in ring allreduce. However, tree allreduce and ring allreduce require repeated gradient aggregations, and compressed gradients cannot be directly summed without first decompressing. Hence, heavy overheads may be incurred. Also, sparsity of aggregated sparse gradients may decrease rapidly during allreduce, and increases communication costs. Moreover, compression after summing up decompressed gradients at intermediate node in an allreduce step incurs further accuracy loss.

"**step sizes are also being rescaled to $\eta_{t-1}/\eta_t$**": Only stepsize for error $e_{t,i}$ is rescaled, not that for $g_{t,i}$.

**Response to Reviewer 3**

"**layerwise compression has been proposed by AdaComp**": Adacomp sparsifies each gradient block by heuristic thresholding and then performs ternary quantization, which requires additional pass to compute maximum gradient values and encode/decoding gradient indices (hefty compression overhead), Moreover, it uses all-to-all broadcast (with heavy network traffic and idle time). The proposed method is a sign-based quantization (easy parallelization and cheap compression) with block-level scaling. It also has provable theoretical guarantees.

"**novelty is limited**": Ours is the first work that studies distributed SGD with Nesterov's momentum and two-way compression. Strong convergence results are provided, showing a linear speedup of using $M$ workers.

"**why [22] is cited?**": That sentence in the introduction is to demonstrate success of deep learning in various applications.

"**cite the following**" Some of them are not very related. Povey et al. (2014) and Chen & Huo (2016) do not consider gradient compression. Chen et al. (2016) is on synchronous SGD, and we have cited the classic papers [6,11,23].

"**all_reduce can be used instead**": Please see our reply to Reviewer 2. Also, we use allreduce for SGDM (line 246).

"**"gradients in a deep network typically have similar magnitudes in each layer"... provide experimental evidence**": Experimental evidence is indeed provided in Figure 1(a) and discussed in lines 181-182.

"**Dist-EF-BlockSGD should be compared with Dist-EF-SGD**": Dist-EF-SGD is a general algorithm that allows arbitrary compressors satisfying Definition 1. Thus, Dist-EF-BlockSGD is a particular instantiation of Dist-EF-SGD. To see the effect of blockwise compressor, we run extra experiments using a ResNet110 on CIFAR-100 with mini-batch size 16 per worker. Dist-EF-BlockSGD improves average test accuracy of non-block version from 74.7% to 75.0%.

"**benefit little from momentum**": This depends on mini-batch size. A larger mini-batch means smaller variance $\sigma^2$, so $(F(x_0) - F_*)$ in the bound of dist-EF-SGDM (lines 211-212) is more dominant. Use of momentum ($\mu > 0$) is then beneficial. Figure 2 shows that with mini-batch size of 16 or 32 (per worker), momentum methods are faster, particularly before epoch 100. At epoch 100, the learning reate is reduced (line 347), and the difference is less obvious.

[Meta-Review · NeurIPS 2019]

The paper introduces an elegant new two-way compression and the use of momentum for such schemes, which are two important questions and as contributions were clearly appreciated by the reviewers, which reached a good consensus. For related work, we comment that in contrast as mentioned, the parallel case of several workers with EF-SGD is considered already in Section 3.5 of https://infoscience.epfl.ch/record/262760/ Also, reviewers were asking if experiments could be expanded to clarify more on how much the momentum aspect improves upon EF alone, and the other detailed comments by the reviewers.